# Biophysical impacts of earth greening can substantially mitigate regional land surface temperature warming

Yitao Li[1,2], Zhao-Liang Li[3] ✉, Hua Wu[1,2], Chenghu Zhou[4], Xiangyang Liu[3], Pei Leng[3], Peng Yang[3], Wenbin Wu[3], Ronglin Tang[1], Guo-Fei Shang[5] & Lingling Ma[6]

Vegetation change can alter surface energy balance and subsequently affect the local climate. This biophysical impact has been well studied for forestation cases, but the sign and magnitude for persistent earth greening remain controversial. Based on long-term remote sensing observations, we quantify the unidirectional impact of vegetation greening on radiometric surface temperature over 2001–2018. Here, we show a global negative temperature response with large spatial and seasonal variability. Snow cover, vegetation greenness, and shortwave radiation are the major driving factors of the temperature sensitivity by regulating the relative dominance of radiative and non-radiative processes. Combined with the observed greening trend, we find a global cooling of −0.018 K/decade, which slows down 4.6 ± 3.2% of the global warming. Regionally, this cooling effect can offset 39.4 ± 13.9% and 19.0 ± 8.2% of the corresponding warming in India and China. These results highlight the necessity of considering this vegetation-related biophysical climate effect when informing local climate adaptation strategies.

According to satellite observations, the earth has been experiencing widespread vegetation greening since the 1980s, primarily due to large-scale climate change and $CO_2$ fertilization effects[1,2]. Such greening could mitigate global warming by triggering negative biochemical feedback to the climate system, which refers to increasing $CO_2$ removal from the atmosphere through the vegetation photosynthesis process[3–5]. Meanwhile, the earth greening could also modify the surface biophysical properties, including the decrease in albedo (enhancing shortwave radiation absorption, known as the radiative process) and the decrease in aerodynamic or surface resistance (enhancing the efficiency of water evaporation or heat convection between the land

surface and atmosphere, known as the non-radiative process), thereby affecting local temperature[6–8]. These biophysical feedbacks could intensify, compensate or even reverse the biochemical force against global warming and thus have drawn much attention in recent years[9–11].

Numerous efforts have been devoted to quantifying the biophysical climate effect of vegetation type conversion, a common situation in land use/land cover changes (LULCC), such as deforestation/afforestation (forests to other vegetation types), wildfire (forests to barren lands) and reclamation (other vegetation to cropland)[12–17]. However, these extreme cases of vegetation type change only occur in specific regions. Analysis of the temperature effect of persistent and

[1]State Key Laboratory of Resources and Environmental Information System, Institute of Geographic Sciences and Natural Resources Research, Chinese Academy of Sciences, 100101 Beijing, China. [2]University of Chinese Academy of Sciences, 100049 Beijing, China. [3]Key Laboratory of Agricultural Remote Sensing, Ministry of Agriculture and Rural Affairs/Institute of Agricultural Resources and Regional Planning, Chinese Academy of Agricultural Sciences, 100081 Beijing, China. [4]Center for Ocean Remote Sensing of Southern Marine Science and Engineering Guangdong Laboratory (Guangzhou), Guangzhou Institute of Geography, Guangdong Academy of Sciences, 510070 Guangzhou, China. [5]School of Land Science and Space Planning, Hebei GEO University, 050031 Shijiazhuang, China. [6]Key Laboratory of Quantitative Remote Sensing Information Technology, Aerospace Information Research Institute, Chinese Academy of Sciences, 100094 Beijing, China. ✉e-mail: lizhaoliang@caas.cn

widespread earth greening can be more constructive for devising better climate mitigation strategies or adaptation policies at different scales.

Remotely sensed observations and earth system models (ESMs) provide tools to explore the climate impact of widespread greening[2]. Due to the uncertainty of underlying physical processes, parameterization schemes and input driving data, models have shortcomings in reproducing the energy partitioning process of vegetation surfaces, thus resulting in biased results[18,19]. Meanwhile, it is difficult to disentangle the unidirectional signal of vegetation greening affecting local climate from the co-evolved satellite vegetation index and temperature observations[20,21]. Hence, previous studies remain debated in terms of the sign and magnitude of temperature responses to the earth greening[20,22].

This study aims to provide solid observational constraints on the biophysical impacts of greening on local temperature. To this end, we assess the potential temperature response to greenness variation across the globe from 2001 to 2018 using satellite-derived land surface temperature (LST) and leaf area index (LAI) as diagnostic variables. Due to the complicated bidirectional effect between vegetation growth and temperature variation, a spatial moving window strategy inspired by the "space-for-time" approach is performed to exclude the impact of long-term climate signals on vegetation growth and acquire the LST sensitivity to LAI[23,24]. Then, the derived LST sensitivity is discussed for different climate conditions and vegetation types at annual and seasonal scales. Moreover, we decompose this sensitivity into contributions from non-radiative, radiative, and indirect climatic feedbacks to further analyze the driving factors behind[6]. Finally, the observed LAI data over the study period is combined with LST sensitivity maps to explore the greening-related climate effect. This estimated signal is subsequently compared with the observed historical temperature variation to evaluate the potential climate benefits of greening at global and regional scales.

## Results

### Biophysical sensitivity of land surface temperature to greening

Figure 1 shows the annual biophysical sensitivity of LST to LAI ($\frac{dLST_{bio}}{dLAI}$) over the study period (2001–2018), which represents the potential annual mean temperature response to one LAI unit increase. Consistent with previous reports of the climate mitigation effect of earth greening[9,25], we find that approximately 71% of the vegetated area shows negative sensitivity, and the global mean value is $-0.46 \pm 1.68$ K m$^{-2}$ (mean ± spatial standard deviation) (Fig. 1a). Vegetation greening in dry and warm regions significantly cools the land surface, and this cooling diminishes and reverses to the warming effect with gradually decreased temperature or increased precipitation (Fig. 1b). Similar to the previous study of the biophysical impact of forestation on LST[26], this climatic variation could be further translated into the latitudinal dependence of a warming effect in northern high latitudes and cooling effects in other latitudes, with the transitional latitude near 50°N (Fig. 1c). However, the difference is found near the equator, where greening only induces a weak cooling effect, but previous studies suggest the strongest cooling of forestation here[27,28]. This difference is mainly due to the inherent difference between the abrupt change from openland to forest and the persistent vegetation greening. Apart from the impact of background climate, we further investigate the $\frac{dLST_{bio}}{dLAI}$ by aggregating all 12 IGBP vegetation types into four broad types (Supplementary Table 1), including forest, other wooden vegetation (OWV), grassland, and cropland (Supplementary Fig. 1). We find strong negative sensitivity in grasslands ($-0.94$ K m$^2$ m$^{-2}$) and

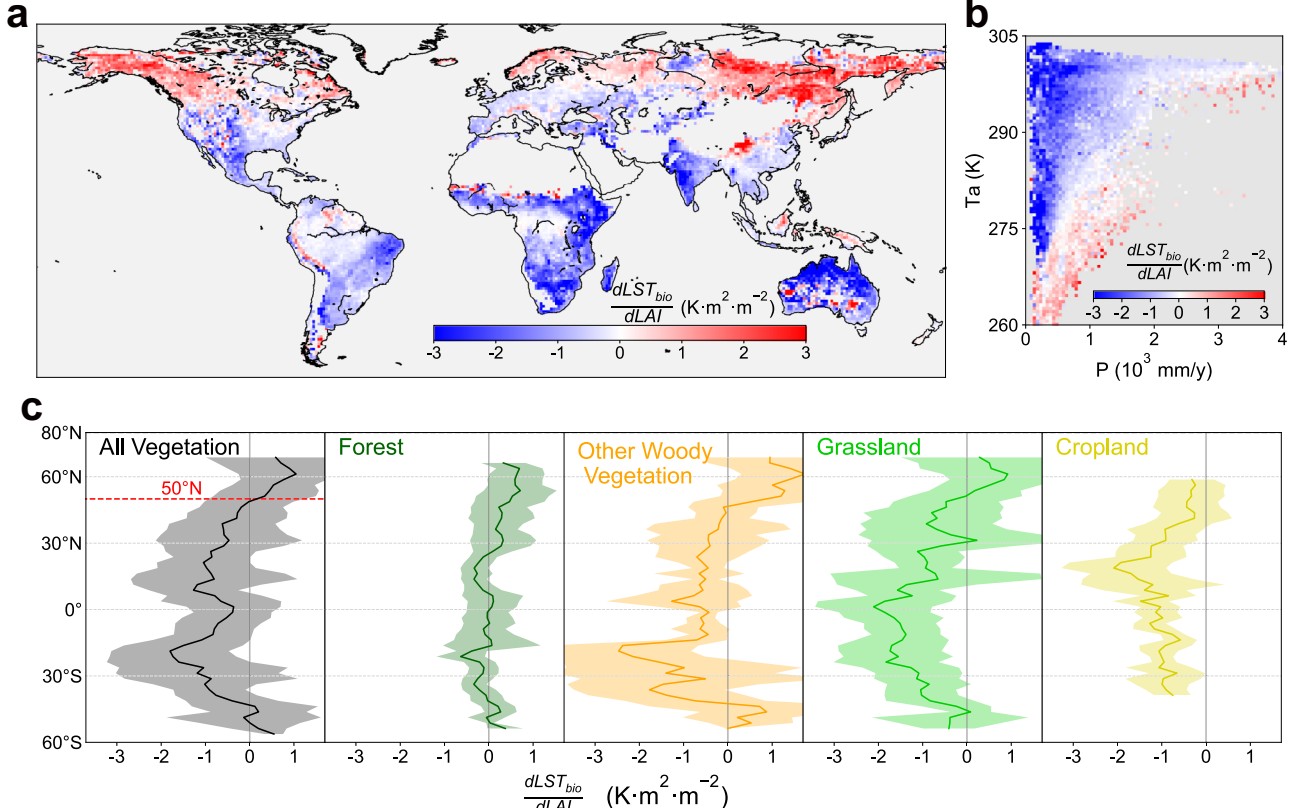

**Fig. 1 | Annual biophysical sensitivity of land surface temperature to leaf area index ($\frac{dLST_{bio}}{dLAI}$) over the study period (2001–2018). a** Spatial map of $\frac{dLST_{bio}}{dLAI}$. **b** Variation in sensitivity means across the climatic bins, in which $\frac{dLST_{bio}}{dLAI}$ is binned as a function of annual precipitation (P, *x*-axis) and air temperature (Ta, *y*-axis). **c** Latitudinal bins of $\frac{dLST_{bio}}{dLAI}$ across different broad vegetation types. The shaded area indicates the latitudinal standard deviation. Source data are provided with this paper.

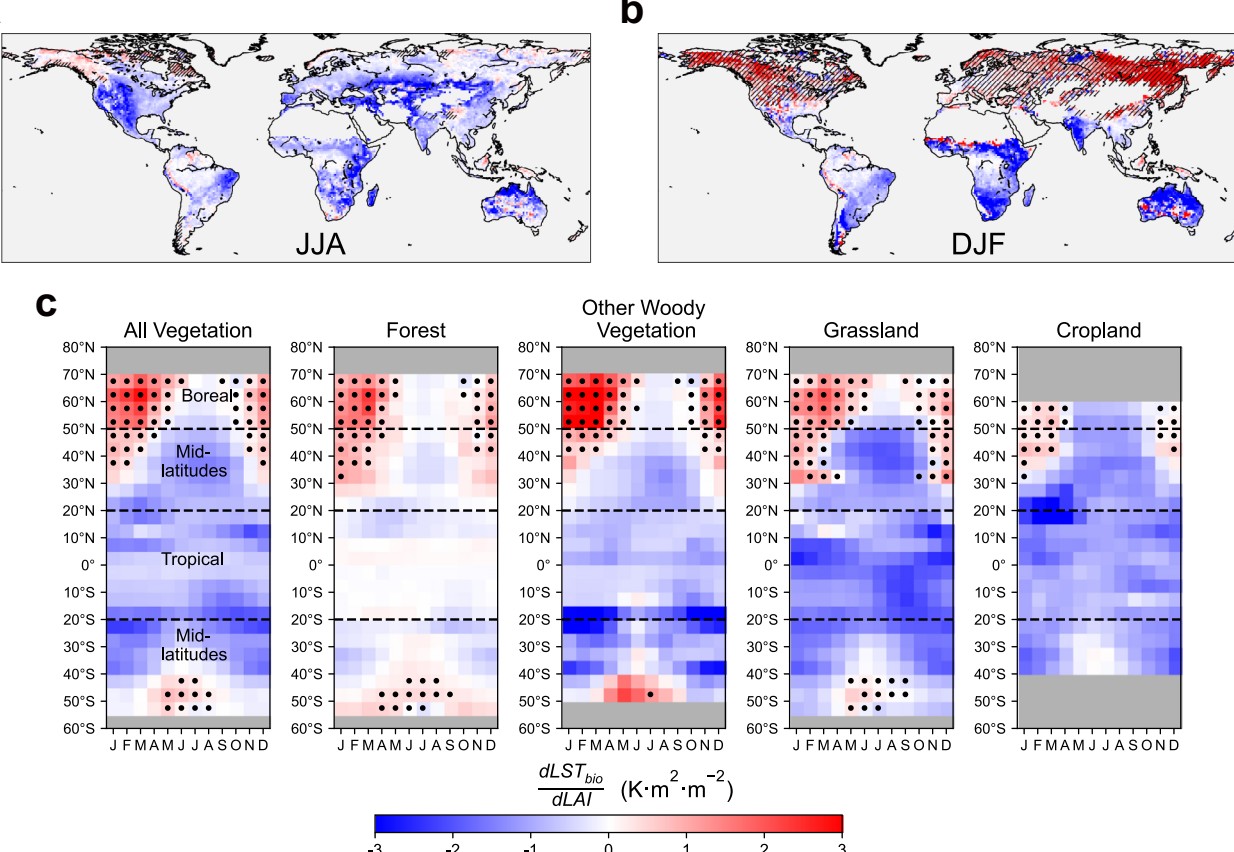

**Fig. 2 | Seasonal pattern of biophysical land surface temperature sensitivity to leaf area index ($\frac{dLST_{bio}}{dLAI}$). a** Spatial map of $\frac{dLST_{bio}}{dLAI}$ in Northern Hemisphere summer (JJA, June to August) and (**b**) in Northern Hemisphere winter (December to February). The shaded area indicates the mean snow cover percentage larger than 1%. **c** Latitudinal means of monthly $\frac{dLST_{bio}}{dLAI}$ for all vegetation, forest, other wooden vegetation, grassland, and cropland. The grids marked with black dots indicate snow coverage that is greater than 1%. Source data are provided with this paper.

croplands (−0.83 K m² m⁻²), then followed by OWV (−0.13 K m² m⁻²), and finally small positive sensitivity of forests (0.16 K m² m⁻²) (Fig. 1c and Supplementary Fig. 2). Overall, compared with the results from the controlled experiments of land surface model, the sensitivities estimated from our observational-based method show similar global magnitude, but with much larger spatial variability[10].

In terms of seasonality, we find larger mean negative sensitivity in Northern Hemisphere summer (JJA, June to August) than in winter (DJF, December to February), with the global mean sensitivity of −0.75 K m² m⁻² and −0.15 K m² m⁻², respectively (Fig. 2a, b). Latitudinally, greening maintains a weak cooling effect throughout the year in tropical regions. In mid-latitudes, greening shows strong growing season cooling and slight cold season warming. With the latitude shifting to polar, the growing season cooling effect shortens, and dormant season warming gradually dominates in boreal regions (Fig. 2c). In addition, we notice that each vegetation type follows this seasonal pattern but with distinctive seasonal magnitude (Fig. 2c). For example, the growing season cooling in grassland and cropland is stronger, and slight growing season cooling is found in forests. Meanwhile, the dormant season warming of OWV and forest is higher than grassland. These different magnitudes of seasonal sensitivity result in the contrasting yearly LST responses for different vegetation types (Fig. 1c).

**Biophysical drivers of LST sensitivity**

The LST sensitivity varies significantly with vegetation type and climatic conditions, but the biophysical mechanism behind is still unclear. Thus, we further decompose the LST sensitivity into the contribution from non-radiative feedback ($\frac{dLST_{bio}^{LE}}{dLAI}+\frac{dLST_{bio}^{H}}{dLAI}$), radiative ($\frac{dLST_{bio}^{\alpha}}{dLAI}$) feedback and indirect climatic ($\frac{dLST_{bio}^{SW\downarrow}}{dLAI}+\frac{dLST_{bio}^{LW\downarrow}}{dLAI}$) feedback based on the energy balance equation (see Methods). Here, we use the same unit as LST sensitivity (K m² m⁻²) to represent the intensity of each process for the convenience of further comparison. Although the decomposition process is performed at a different scale, the results show that the sum of all contributors agrees well with the directly calculated LST sensitivity (Supplementary Fig. 3, 4, Supplementary Discussion 1). As shown in Fig. 3a, we find a minor impact from indirect climatic feedback (yellow line) for all latitudes, suggesting the local temperature effect of vegetation greening is mainly induced by the direct modification of surface biophysical parameters. In boreal regions (>50°N), radiative warming (red line) surpasses non-radiative cooling (blue line), resulting in a positive LST signal (black line). Monthly results further indicate that this positive signal mainly occurs from January to April, with the maximum value in March (Fig. 3b). This seasonal variation is the combined result of albedo sensitivity to LAI and incoming shortwave radiation (see "Methods"). For the remaining latitude zones, our results show that non-radiative cooling offsets radiative warming and dominates the final negative LST sensitivity (Fig. 3a). Symmetrical latitudinal patterns are found between the radiative warming and non-radiative cooling, which suggests that their intensity may be controlled by the same factors. Seasonally, the non-radiative cooling shows larger magnitudes in the growing season than in the dormant season (Fig. 3c, e), leading to the seasonal pattern of LST sensitivity in mid-latitudes (Fig. 2c). However, no significant

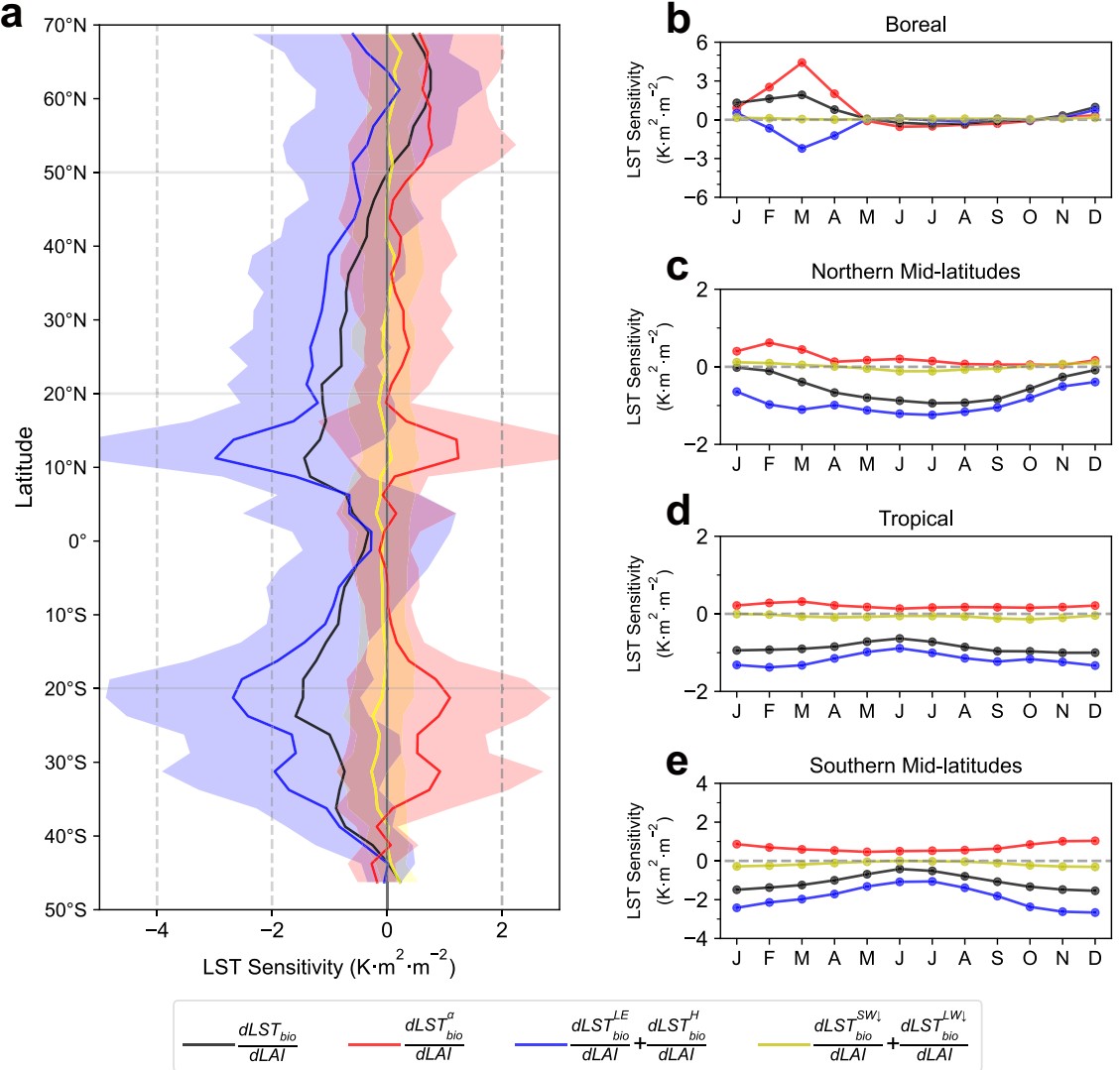

**Fig. 3 | The decomposition of land surface temperature (LST) sensitivity to leaf area index. a** The latitudinal pattern of final LST sensitivity ($\frac{dLST_{bio}}{dLAI}$, black line) and equivalent LST sensitivity from different biophysical processes, including radiative ($\frac{dLST_{bio}^{LE}}{dLAI}$, red line), non-radiative ($\frac{dLST_{bio}^{LE}}{dLAI}+\frac{dLST_{bio}^{H}}{dLAI}$, blue line) and indirect climate ($\frac{dLST_{bio}^{SW\downarrow}}{dLAI}+\frac{dLST_{bio}^{LW\downarrow}}{dLAI}$, yellow line) feedback. The shaded area indicates the latitudinal standard deviation. **b** Seasonal pattern of LST sensitivity and mentioned biophysical processes in boreal, (**c**) northern mid-latitude, (**d**) tropical, and (**e**) southern mid-latitude regions. Source data are provided with this paper.

seasonal variation is found for the two processes or LST sensitivity in the tropics (Fig. 3d).

In the boreal cold season, the dominant role of the radiative process can be attributed to the presence of snow[17,29], which is confirmed by the co-occurrence of snow and positive sensitivity in monthly results (Fig. 2). Theoretically, the dense vegetation canopy is more likely to cover the snow background than the sparse canopy; thus, the albedo difference is intensified between vegetated surfaces with different LAI values in snowy weather, leading to the stronger radiative warming effect of the higher LAI surface[30–32]. This biophysical impact of snow in boreal regions is further confirmed by the positive linear relationship between mean monthly snow coverage and mean monthly LST sensitivity from January to April (Fig. 4a), when the LST sensitivity is found positive (Fig. 3b). As expected, this impact of snow can be traced back to the positive linear relationship between snow coverage and the radiative process (Fig. 4b), but the slope is larger than that in Fig. 4a (0.032 vs. 0.023). This result suggests that only about three-quarters of the energy absorption induced by the snow masking effect is used for boosting the LST, and the remaining quarter is compensated by the energy diffusion from the land surface to the

atmosphere (Fig. 4c), which is confirmed by the negative slope between the non-radiative process and snow coverage (−0.009). These results provide evidence to support the dominant role of snow in controlling the boreal positive LST sensitivity, whose impact is underestimated in previous model-based studies[10,11].

For the remaining snow-free regions, our analysis reveals the strong control of vegetation greenness in modulating the magnitude of negative LST sensitivity. Specifically, an exponential function is used to characterize the observed relationship between LAI and LST sensitivity (Fig. 4d). This relationship suggests that the greening of low LAI areas could induce a stronger cooling effect, and this cooling effect gradually decreases and converges to zero with the increase in LAI. This exponential relationship could be traced back to similar results in radiative and non-radiative processes with contrasting temperature effects (Fig. 4e, f). In the competition between the two biophysical processes, non-radiative cooling takes the dominant position and results in negative LST sensitivity. However, the intensities of both processes are larger in lower greenness areas and decrease as the LAI turns greener. The mechanism is that biophysical parameters, such as albedo and roughness length, gradually saturate with the LAI increase,

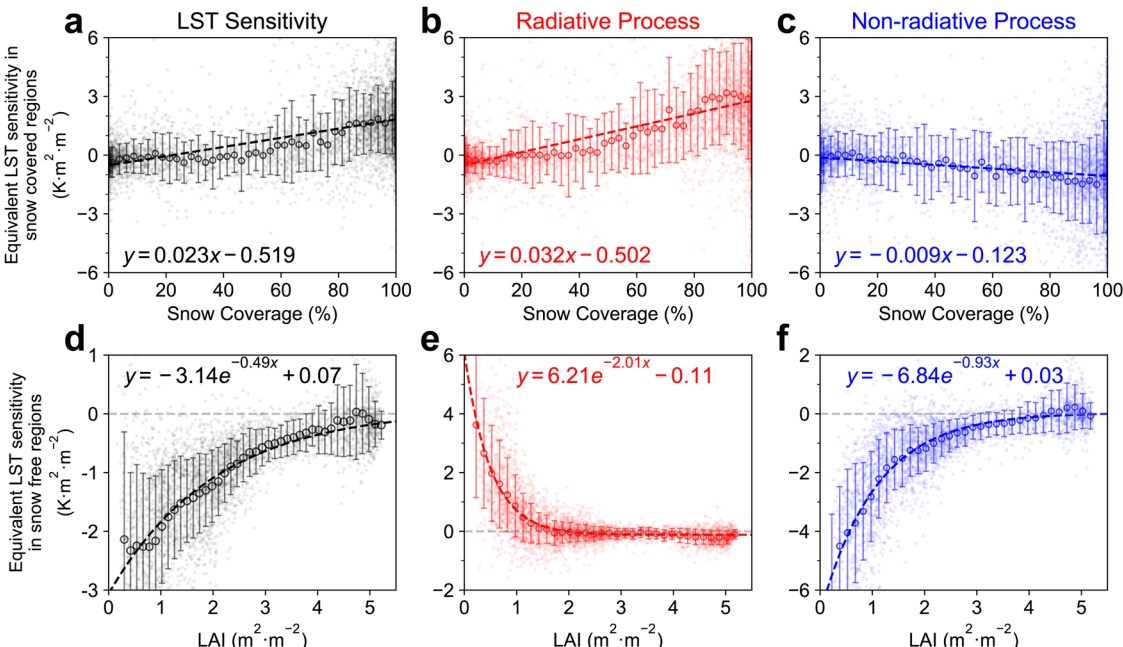

**Fig. 4 | Impacts of snow coverage and leaf area index (LAI) on the local bio-physical feedbacks. a** The relationship between mean snow coverage and land surface temperature (LST) sensitivity and the equivalent LST sensitivity from (**b**) radiative process and (**c**) non-radiative process in snow-covered area (snow coverage that is greater than 1%) from January to April. Error bars show the standard deviation of the sensitivity within the snow coverage bin (±2%). **d** The relationship between annual LAI and LST sensitivity and the equivalent LST sensitivity from (**e**) radiative process and (**f**) non-radiative process in snow-free regions. Error bars show the standard deviation of the sensitivity within the LAI bin (±0.15 m² m⁻²). Source data are provided with this paper.

leading to the weak magnitude of biophysical feedbacks in high greenness areas[33,34]. In addition, we notice that the radiative warming effect is almost zero when LAI reaches 1.5 m² m⁻², and reversed but slightly radiative cooling could be found in higher greenness areas (Fig. 4e). A possible explanation is that canopy development in such high LAI regions could intensify the near-infrared (NIR) reflection[35], leading to the less energy absorption. Overall, our inference of LAI control could explain the large variability in LST sensitivity across different vegetation types and climate zones (except for cold climates) (Supplementary Fig. 2). For example, grassland and cropland (lower LAI) have stronger cooling effects, and forest (higher LAI) shows no significant temperature sensitivity in temperate and equatorial climates (Supplementary Fig. 5). Moreover, the extensive cooling effect in arid regions could be also partly attributed to the lower LAI value of vegetation in water-limited environments (Fig. 1b).

Interestingly, we find that the relationship between LAI and LST sensitivity is not universal. The fitting results of snow-free regions in different latitude zones show larger negative sensitivity in the tropics than in mid-latitudes for the same LAI value (Supplementary Fig. 6). This difference is most likely induced by the inherent climatic condition differences at different latitudes. Considering that downward shortwave radiation is the driving force of most latitudinal phenomena, we further investigated its potential impact on LST sensitivity. The binned means of LST sensitivity against LAI and downward shortwave radiation indicate a stronger negative LST sensitivity under the high incoming radiation and low LAI value conditions (Supplementary Fig. 7a). Radiative and non-radiative feedbacks are also intensified as the downward shortwave radiation increases or LAI decreases (Supplementary Fig. 7b, c). As the radiative forcing is relatively strong in the tropics, the impact of shortwave radiation could explain the difference in Supplementary Fig. 6. Moreover, we infer that this climatic control can explain the seasonal pattern of $\frac{dLST_{bio}}{dLAI}$ in mid-latitudes, where LST sensitivity in the growing season (abundant solar radiation) is higher than that in the dormant season (limited solar radiation) (Fig. 3c, e).

## Climate benefit of greening over the last 20 years

The LST sensitivity represents only the temperature response of the ideal average greening scenario worldwide. Here, we further quantify the actual LST effect based on the sensitivity map and the observed LAI trend over the study period. Results show a widespread greening trend with a global mean of 0.037 m² m⁻² decade⁻¹ (Supplementary Fig. 8). Correspondingly, the global mean greening-induced temperature trend ($\delta LST_{bio}$) is −0.013 ± 0.009 K/decade (Fig. 5a), which offsets 4.6 ± 3.2 % of the warming trend of 0.289 K/decade within the vegetated area (Supplementary Fig. 9). When considering those pixels with statistically significant LAI trends (Mann-Kendall test, $P < 0.05$), the cooling trend can increase to −0.029 ± 0.008 K/decade. Greening in such regions could offset about 9.2 ± 3.7% of the corresponding warming trend, and contribute about 68% of the global cooling signal. As shown in Fig. 5b, we confirm the greening-induced acceleration of global warming in the boreal region (0.008 ± 0.007 K/decade), which is primarily affected by the substantial warming trend in winter (0.020 ± 0.006 K/decade) and spring (0.022 ± 0.009 K/decade). For the remaining latitudinal zones, the annual mean temperature signal of −0.020 ± 0.012 K/decade could offset approximately 5.9 ± 3.8% of the warming trend, which is mostly contributed by the cooling from 0° to 50°N (Fig. 5b). The cooling trend shows dramatic seasonal variation in northern mid-latitudes (−0.075 ± 0.027 K/decade in summer, −0.010 ± 0.004 K/decade in winter) (Fig. 5b), which may reduce the intra-annual temperature variation. Meanwhile, the significant cooling in the Northern Hemisphere summer can also reduce the risk of heatwaves in the future with more frequent extreme weather[36]. Overall, our result of the global magnitude of greening-related cooling is comparable with former model-based studies but with larger spatial or seasonal variability[10,11,37].

We further rank the annual climate impact of vegetation greening at the biome level (Fig. 5c). The results show that the greening of croplands could induce substantial cooling (−0.066 ± 0.016 K/decade, rank 4), especially in the arid (−0.112 ± 0.034 K/decade, rank 1) and dry equatorial climate (−0.105 ± 0.029 K/decade, rank 2). This dramatic

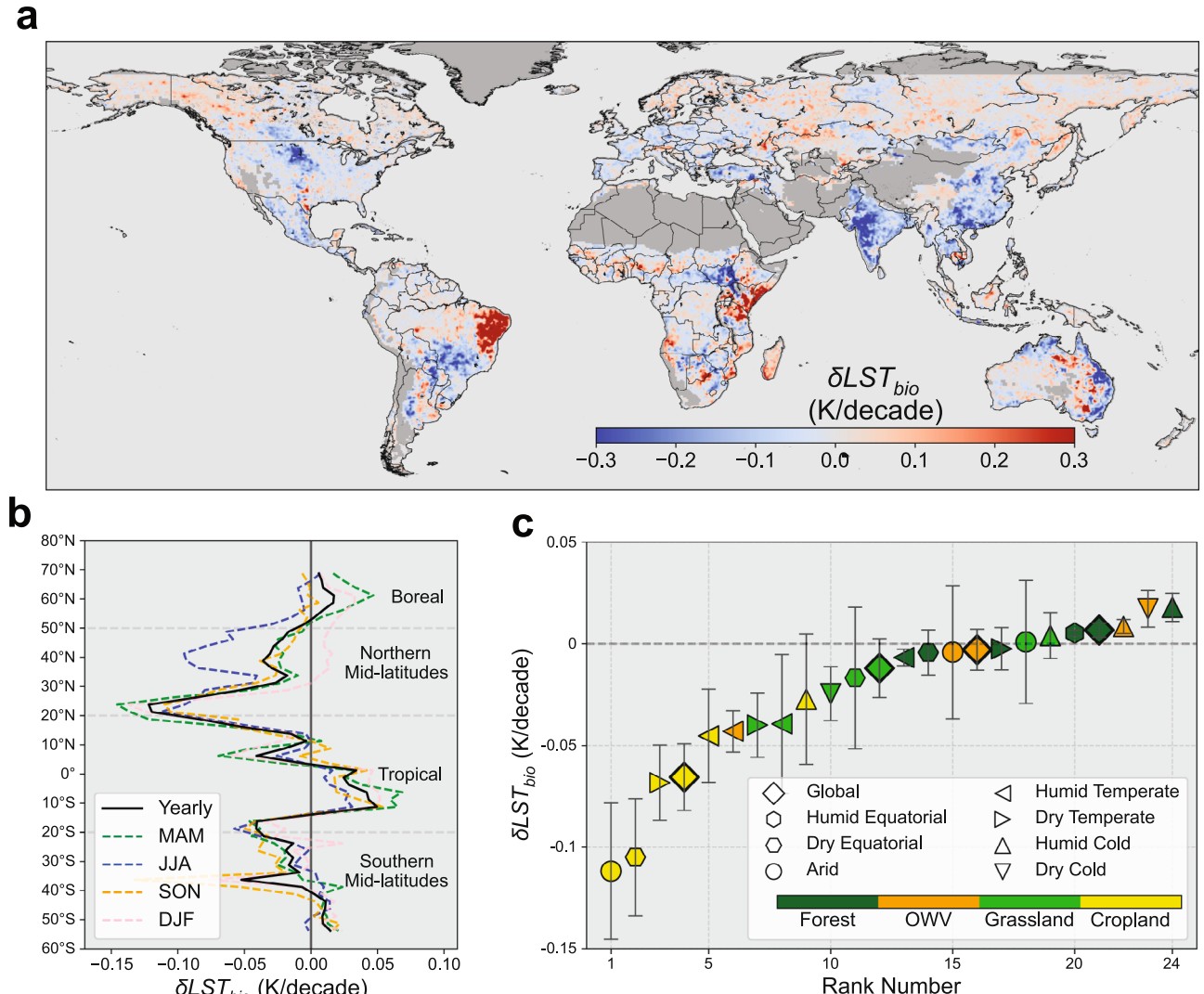

**Fig. 5 | The land surface temperature trend induced by biophysical feedback of earth greening ($\delta LST_{bio}$) over 2001–2018. a** Global map of $\delta LST_{bio}$. **b** Latitudinal means of annual and seasonal $\delta LST_{bio}$. Abbreviations: MAM, March to May; JJA, June to August; SON, September to November; DJF, December to February. **c** Ranking of mean $\delta LST_{bio}$ for different biomes. The error bars represent the 95% confidence interval of the temperature trend. Only those biomes with the area greater than 1.5% of the global vegetation area are shown. OWV other woody vegetation.

cooling of global croplands should be the result of both large LST sensitivity (−0.83 K m$^2$ m$^{-2}$) and LAI trend (0.072 m$^2$ m$^{-2}$/decade). Nevertheless, the greening of grasslands with higher LST sensitivity (−0.94 K m$^2$ m$^{-2}$) only induces slight global cooling (−0.012 ± 0.014 K/decade, rank 12), mainly due to the lower LAI trend (0.031 m$^2$ m$^{-2}$/decade) and the presence of grasslands in the humid cold climate, where greening could induce the opposite warming effect (0.004 ± 0.0011 K/decade, rank 19). Conversely, the greening of forests could accelerate global warming by 0.007 ± 0.003 K/decade. Since the LST sensitivity is small in high LAI areas (Fig. 4d), the cooling effect of mid- and low-latitude forest greening is very limited. This global positive trend is primarily contributed by the warming trend of forests in humid cold regions (0.018 ± 0.007 K/decade, rank 24), where snowfall amplifies the positive LST sensitivity (Fig. 5c). However, this result only represents the warming effect of forest greening through the biophysical feedback, and the further assessment of net climate effect should also take the biochemical feedback into account[4,28].

To investigate the regional climate benefit of greening, we calculate the mitigation ratio of greening using the observed surface air temperature trend ($\delta T_{obs}$) and the estimated greening-induced temperature trend ($\delta LST_{bio}$) at pixel and national scale (see "Methods").

The higher value of this ratio indicates larger warming trend the biophysical feedback of greening can offset. As shown in Fig. 6a, regions where greening can significantly mitigate climate change are overlapped with significant greening areas, (e.g., China, India, Europe and southern Brazil). These regions have also been confirmed by the previous study to dominate the global greening signal after the 21$^{st}$ century[38]. We further compare $\delta LST_{bio}$ and $\delta T_{obs}$ in 10 countries with sizable vegetated areas (defined as vegetation areas larger than 1.5 million km$^2$). As shown in Fig. 6b, India and China are ranked as the top 2, where substantial greening induces strong cooling effects of −0.140 ± 0.048 K/decade and −0.059 ± 0.009 K/decade. These cooling effects can offset 39.4 ± 13.9% and 19.0 ± 8.2% of the surface warming, respectively. Australia experiencing a lower greening trend also shows intense greening-related cooling, but with significant uncertainty (−0.030 ± 0.071 K/decade). These results emphasize the important role of greening's biophysical feedback for regional climate mitigation. For the countries that rank fourth to eighth, the cooling effects of greening are small, and only European Union (EU) shows significant greening-related mitigation. In contrast, our results indicate that greening in Canada and Russia has slightly accelerated warming trends, mainly because of their positive LST sensitivity (Fig. 1a). Brazil

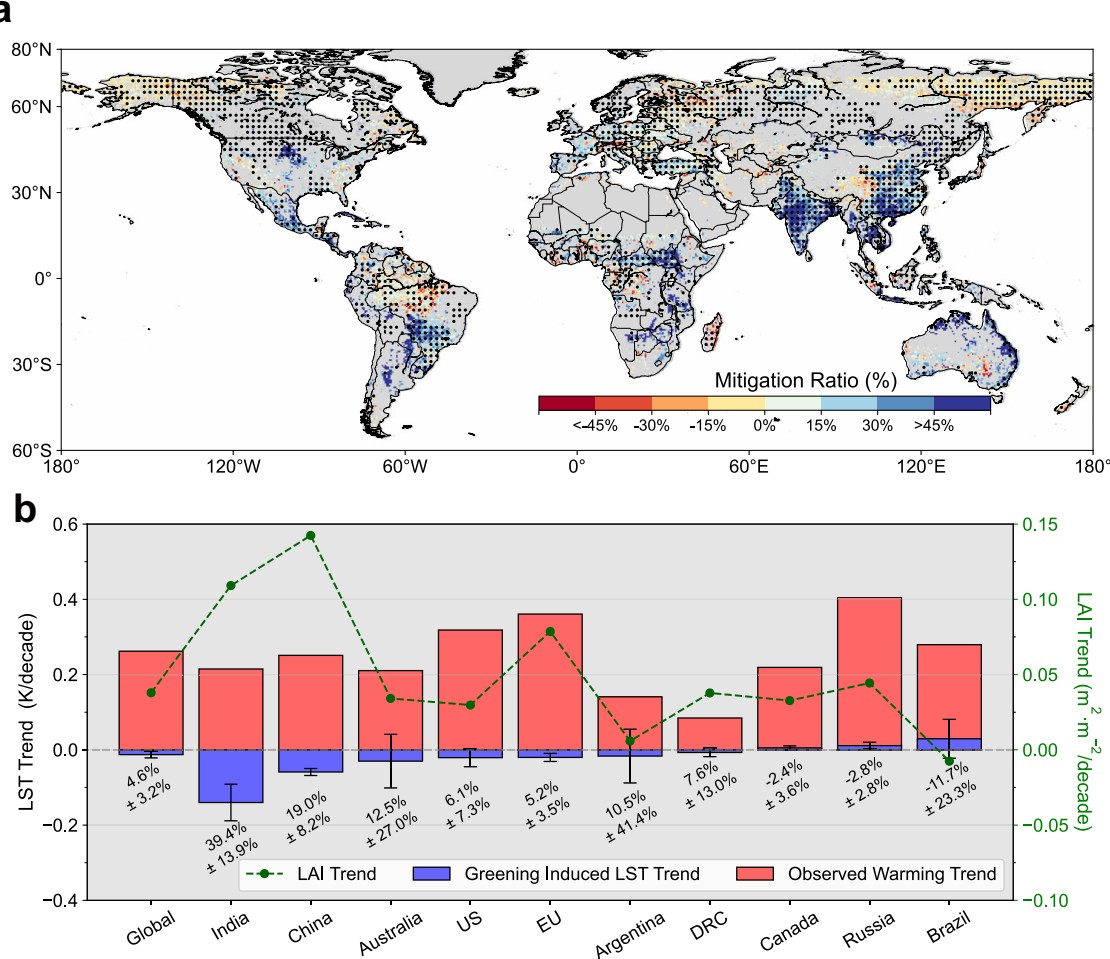

**Fig. 6 | Potential mitigation effect of biophysical impact of earth greening.**
**a** Spatial map of mitigation ratio. Only pixels that are significant at 95% confidence interval are shown. Areas with statistically significant greening trends are masked by black dots (Mann–Kendall test, $P < 0.05$). **b** Comparison of observed temperature trends and greening-induced land surface temperature trends in 10 sizeable vegetated countries. Error bars indicate the uncertainty of greening induced temperature trend. The percentage under each blue bar indicates the climate mitigation ratio (mean ± uncertainty). Here, European Union (EU) is included in the analysis. US United States, DRC Democratic Republic of the Congo. Source data are provided with this paper.

also shows the warming associated with vegetation change, but the reason is the browning signals in the rainforests and Cerrado. At the same time, the cooling due to the greening of pastures and agricultural land partially compensates for this warming effect (Fig. 6 and Supplementary Fig. 8), and thus lead to large uncertainty of the temperature trend.

## Discussion

Using satellite observations, we investigate the biophysical impact of earth greening on LST from 2001 to 2018 at a finer scale. The spatial regression model developed in our study helps us disentangle the unidirectional LAI-LST effect from the simultaneously changing climate and vegetation greening signal (Supplementary Discussion 2). Our results provide evidence to support the greening-induced warming effect in boreal regions (Fig. 5), which has still been under debate in recent years[9,10,20]. Moreover, by further looking into the different biophysical processes and their driving factors, our study could improve the understanding of the biophysical feedback mechanisms of widespread earth greening. The deeper meaning of knowing these control factors is to enable reasonable predictions of future temperature sensitivity. We believe in the potential climate mitigation from greening in boreal regions in a projected warming world, mainly because the decreased snow cover will weaken the radiative warming in the future[29] (Fig. 4a). On the other hand, our view of LAI controlling the biophysical

sensitivity suggests the decline in non-radiative cooling in the future, as the turbulence fluxes will be less sensitive in a continuously greening world due to the saturation effect (Fig. 4d–f). However, a previous study suggests the increased sensitivity of turbulence fluxes to vegetation greenness driven by climate change[18]. Given the different views from previous studies and the tradeoff between the two mentioned mechanisms, we argue that the variation in future biophysical LST sensitivity remains uncertain and requires further quantitative assessment.

The increasing $CO_2$ concentration is the major driving force of both global warming and earth greening in recent decades. However, compared with the global warming rate, we confirm the very limited effect of biophysical-based climate mitigation from earth greening. Meanwhile, our results also provide evidence of the significant surface cooling in regions with extensive greening trends. Specifically, we find that greening-related cooling can offset about 20% and 40% of the warming trend in China and India (Fig. 6b), respectively. These results highlight the role of the biophysical impact of greening in future adaptation strategies against ongoing warming. A previous study proved that China and India are the leaders in global greening in the 21st century, which is achieved by human land-use management, such as the afforestation project in China[8] and the increased harvested area by fertilization and irrigation in India[38]. This suggests the large potential of human land use and land management strategies and

ecological projects to mitigate climate pressure, not only through carbon uptake from the atmosphere but also the biophysical processes. Nevertheless, afforestation engineering is also performed in the EU[39], and the greening trend is also significant here, but a very limited climate impact from the biophysical feedback of greening is observed (Fig. 6b). The contrasting results suggest the importance of prioritizing the geolocation of ecological projects to reach the maximum climate benefit. To this aim, our assessment of global LST sensitivity could be very helpful for informing regional vegetation-based climate adaptation policies for the present and future[28].

This study conducts controlled experiments through the space-for-time analogy to extract the sensitivity of surface temperature to vegetation change at a finer scale, and we reveal the climate impacts of earth greening with substantial spatial heterogeneity and seasonal variability. However, there are still caveats of our study that should be highlighted due to the limitations of the data and methods. First, our mapped sensitivity may have relatively large uncertainty in the low greenness area, which is reflected in the large standard deviation of scatters with low LAI in Fig. 4d. In such regions, the regressed slope could be sensitive to the noise of input data, as the difference among input LAI is relatively small. However, we use an additional filter to restrict the minimal LAI difference to ensure the robustness of our results. Meanwhile, this sensitivity uncertainty is also reflected in the uncertainty of the temperature effect ($\delta LST_{bio}$) after combining with the observed LAI changes (see Methods). Second, our analysis quantifies only the local temperature signal of greening but ignores the further impact of large-scale feedback on climate, which is difficult to capture in data-driven studies[40,41]. Notably, this indirect climate impact is non-local or teleconnected and determined by the scale and geolocation of surface changes. Nevertheless, our result could still be useful for local climate adapting strategies and for the comparison or evaluation of simulated sensitivity from land surface models, as these models only focus on the land surface process without the atmospheric circulation process involved.

In addition, we adopt satellite-derived LST as the indicator of surface thermal conditions but not near-surface air temperature, although the latter might be more relevant to human living conditions. On the one hand, we use LST because remote sensing observations provide direct measurements of radiometric surface temperature in terms of data availability. The global estimation of near-surface air temperature generally requires complex statistical models or assimilation methods combined with site observations, and thus, the data uncertainty may be substantial in such regions with the sparse spatial distribution of observation sites. On the other hand, we argue that the usage of LST can better reflect the pure climate impact from land surface greening for our method because the variation in near-surface air temperature is a hybrid signal from atmospheric and land surface processes, thus, the sensitivity derived from the air temperature may underestimate the actual climate impact due to horizontal heat convection in the near-surface atmosphere. However, when the acquisition of sensitivity is not limited to our method, previous studies have also shown the different responses of air temperature and surface temperature to vegetation changes[42,43]. Specifically, based on the ESM simulations of different scenarios, the local 2 m air temperature sensitivity is about 35% to 65% of the local surface temperature sensitivity. If we take this difference in temperature measurements into account, the cooling effect of greening will be also correspondingly halved. However, the climate mitigation ratio could reach 24.5% and 10.5% in India and China (assuming the air temperature sensitivity is half the surface temperature sensitivity), which still shows the strong climate mitigation effects of anthropogenic greening. Thus, we argue the role of vegetation greening cannot be ignored in the assessment of future climate, especially for the hot spots of greening induced directly by human activities.

## Methods

### Biophysical and environmental parameters

The data for modeling the biophysical sensitivity include surface biophysical properties from MODIS (Moderate-resolution Imaging Spectroradiometer) observations and other environmental variables at the monthly scale.

The monthly mean land surface temperature (LST) over the study period (2001–2018) is generated through a two-step strategy. The original LST data are from the collection 6 MOD11A1 and MYD11A1 products with a 1-km spatial resolution downloaded from the National Aeronautics and Space Administration (NASA) website. Based on the quality control flag, only data with LST errors less than 2 K are used. First, the instantaneous temperature observations are converted to the daily mean temperature using a statistical-based method[44]. This method generates the daily mean LST for each pixel using the weighted mean value of different MODIS observation combinations (at least one daytime and one nighttime observation for both Aqua and Terra observations). Second, the daily average temperature is temporally aggregated to the monthly value using averaged by day (ABD) approach[45] and spatially aggregated to 0.05° resolution. The flow chart for the monthly LST production can be found in Supplementary Fig. 11. Compared with the direct use of the arithmetic mean of day and night LST observations from the MYD11C3 product (MODIS/Aqua monthly 0.05°), the two-step strategy could significantly enhance the data availability by combining Aqua and Terra observations (Supplementary Fig. 12), which is important for further analysis.

Leaf area index (LAI) is used to characterize the structural properties and vegetation greenness. The major LAI data from 2001 to 2018 used in this study are from the Global Land Surface Satellite (GLASS) dataset, which is based on MODIS surface reflectance observations and generated by general regression neural networks (GRNNs)[46]. The GLASS LAI product has 0.05° spatial resolution and 8-day temporal resolution, with wall-to-wall spatial coverage from 70°N to 70°S. The 8-day data are temporally aggregated to the monthly scale using the mean value. In addition, alternative GLOBMAP (0.072°, 8-days) and GIMMS3g (0.083°, 15-days) LAI datasets[47,48] are also resampled and temporally aggregated to the same resolution (0.05°, monthly) to further test the robustness of the results (Supplementary Discussion 3).

Annual maps of dominant land cover types and the subgrid frequency over 2001–2018 are from the MODIS collection 6 land cover product MCD12C1, with a spatial resolution of 0.05°[49]. The land cover maps used in our study are based on the International Geosphere-Biosphere Programme (IGBP) classification scheme. In this study, only regions covered by vegetation are examined, and the broad vegetation classes are aggregated from all IGBP vegetation types for further analysis[10] (Supplementary Table 1 and Supplementary Fig. 2).

The climate variables from 2001–2018 used in our analysis include precipitation, air temperature and radiation data from the ERA5-Land monthly averaged reanalysis data with a 0.1° spatial resolution. Precipitation and air temperature are used to study the influence of background climate on the biophysical feedback intensity. Radiation data, including surface downward solar radiation (SW↓) and surface downward longwave radiation (LW↓), serve as the attribution factors in further analysis.

Turbulence fluxes, including latent heat (LE) and sensible heat (H), are crucial biophysical parameters involved in our attribution analysis. In this paper, LE and H data are from the FluxCom dataset[50], which is derived from the combination of flux tower measurements, remote sensing data, and meteorological data using model tree ensemble (MTE) regression. The original FluxCom dataset has 1/12° spatial and monthly temporal resolutions. We resample the original data to a 0.1° resolution to match the radiation data. Notably, turbulent heat fluxes data used in this study are from 2001 to 2015, as the dataset is only updated to 2015.

Monthly ground heat flux (G) data over 2001–2015 are from outputs of NOAH land surface model in Global Land Data Assimilation System 2.1 (GLDAS-2.1) dataset. GLDAS dataset with the spatial resolution of 0.25° is resampled into 0.1° to match other energy balance terms data.

Albedo (α) data over 2001–2015 are extracted from the MODIS collection 6 product MCD43C3, which has the same spatial resolution of 0.05° and 16-day temporal resolution. Due to the strong correlation and small difference between black- and white-sky albedo, we simply calculate the actual albedo using the arithmetic average of black- and white-sky albedo for each pixel[17]. The monthly albedo data are temporally aggregated from the original data using the mean value and then spatially aggregated to a 0.1° spatial resolution.

The digital elevation model (DEM) with a 0.05° spatial resolution is used to filter out the impact of altitude on the local temperature, which is spatially aggregated from the version 4 shuttle radar topography mission (SRTM) with a 90 m resolution.

The monthly snow cover data (MYD10CM) from the National Snow & Ice Data Center (NSIDC) at a 0.05° spatial resolution from 2001–2018 are used to analyze the impact of snow on the biophysical feedback.

### Estimation of the biophysical LST sensitivity to the LAI

As complex bidirectional interactions exist between the local background climate and vegetation growth conditions on different scales[9,20,22], we apply a moving window strategy inspired by the "space-for-time" method to calculate the biophysical sensitivity of LST to LAI ($\frac{dLST_{bio}}{dLAI}$) in this study. The "space-for-time" method has been widely used to investigate the local temperature impact of LULCC. It assumes that the target pixel shares the same background climate with adjacent pixels within the moving window, so the LST difference in the target and contrasting pixels is induced by the biophysical feedbacks of land cover change[23,24,51]. Similarly, we assume vegetation greenness can be the only driving factor of LST spatial variation under certain restrictions, the biophysical sensitivity of LST to LAI could be regressed from spatial nearby LAI and LST observations. The advantage of this method over the temporal regression strategy[9] is the exclusion of the impact of climate natural variability or the long-term warming trend on vegetation growth, because pixels with different LAI values within the moving window share the same background climate (Supplementary Fig. 16).

This spatially moving window strategy is used to produce monthly $\frac{dLST_{bio}}{dLAI}$ over the study period. The specific way of the strategy works is as follows. For a given target pixel, all the potential samples for comparison are from the spatial nearby pixels within the moving window, which is set to 50 × 50 km (9 × 9 pixels at the equator) according to the previous studies[8,52]. We further set two screening criteria for all candidates to exclude the influence of land cover and elevation difference: (1) the selected pixel should have the same main land cover type as the target pixel, and the coverage percentage difference should be less than 10% according to MODIS landcover data; (2) the elevation difference between the selected and target pixels should be less than 100 m. Then, we can obtain the biophysical sensitivity for the target pixel through the regression of LAI and LST differences between all selected comparison pixels and the target pixel. Here, the nonparametric Theil-Sen's slope is used to solve the potential skewed distribution problem of the samples[53]:

$$slope = median\left(\frac{y_i - y_j}{x_i - x_j}\right) \quad (1)$$

here $x$ and $y$ indicate the LAI and LST differences; $i$ and $j$ are the geolocations of samples within the moving window. Theil−Sen slope estimator adopts the median value of a range of possible slopes and is thus insensitive to the statistical outliers of the samples. In addition, we only calculate the sensitivity if there are at least four valid samples, and with a minimum LAI difference larger than 0.1 m² m⁻² to further ensure the robustness of our result. Here, a positive sensitivity value means that vegetation greening has a warming effect on the local climate and vice versa.

### Decomposition of the biophysical LST sensitivity

To analyze the contribution of different biophysical feedbacks to the final LST sensitivity, we perform a decomposing procedure based on the energy balance equation:[54,55]

$$SW \downarrow (1 - \alpha) + \varepsilon LW \downarrow - \varepsilon\sigma(LST)^4 = H + LE + G \quad (2)$$

where $SW \downarrow$ and $LW \downarrow$ are shortwave and longwave downward radiation, $\alpha$ indicates albedo; $\sigma$ denotes the Stephan-Boltzmann constant ($5.67 \times 10^{-8}$ W m⁻² K⁻⁴); $\varepsilon$ indicates surface emissivity, which is estimated from the empirical relationship with albedo for vegetated surfaces ($\varepsilon = 0.99\text{-}0.16\alpha$);[55] H, LE, and G are respectively sensible heat, latent heat, and ground heat flux. Notably, energy balance closure is required for the decomposition model[56], but the energy balance terms used in this study are from different datasets and thus are not closed. To account for this, we then perform the energy residual distribution using the Bowen ratio method, which assumes the ratio of H to LE is invariant[50].

$$LE_{corrected} = (R_n - G) \times \frac{LE}{LE + H} \quad (3)$$

$$H_{corrected} = (R_n - G) \times \frac{H}{LE + H} \quad (4)$$

$$R_n = SW \downarrow (1 - \alpha) + \varepsilon LW \downarrow - \varepsilon\sigma(LST)^4 \quad (5)$$

Notably, this invariant assumption does not mean the Bowen ratio is constant when the vegetation has changed. Subsequently, the corrected turbulent fluxes ($LE_{corrected}$ and $H_{corrected}$) data are used for further attribution analysis in this paper.

The monthly LST sensitivity, which is represented by the upward longwave radiation through the Stephan-Boltzmann expression[54], could be divided into the contribution of each term using the first-order Taylor expansion:

$$\frac{dLST_{bio}}{dLAI} = \frac{dLST_{bio}^{\alpha}}{dLAI} + \frac{dLST_{bio}^{LE}}{dLAI} + \frac{dLST_{bio}^{H}}{dLAI} + \frac{dLST_{bio}^{SW\downarrow}}{dLAI} + \frac{dLST_{bio}^{LW\downarrow}}{dLAI} \quad (6)$$

where the five terms on the right side of the equation denote the equivalent LST sensitivity from the corresponding biophysical pathway:

$$\frac{dLST_{bio}^{\alpha}}{dLAI} = -\lambda_0 \frac{d\alpha_{bio}}{dLAI} SW \downarrow \quad (7)$$

$$\frac{dLST_{bio}^{LE}}{dLAI} = -\lambda_0 \frac{dLE_{bio}}{dLAI} \quad (8)$$

$$\frac{dLST_{bio}^{H}}{dLAI} = -\lambda_0 \frac{dH_{bio}}{dLAI} \quad (9)$$

$$\frac{dLST_{bio}^{SW\downarrow}}{dLAI} = \lambda_0 \frac{dSW \downarrow_{bio}}{dLAI}(1 - \alpha) \quad (10)$$

$$\frac{dLST_{bio}^{LW\downarrow}}{dLAI} = \lambda_0 \left(\frac{dLW \downarrow_{bio}}{dLAI} \varepsilon\right) \quad (11)$$

Here, $\lambda_0$ is given by the first-order Taylor expansion term of upward longwave radiation:

$$\lambda_0 = \frac{1}{4\varepsilon\sigma(LST)^3} \qquad (12)$$

Note that the sensitivity terms related to emissivity and ground heat flux are ignored, as their effects are relatively small[37,52,55]. Meanwhile, we neglect the complex effect of LST on other energy balance terms in the attribution model for the following two reasons: (1) attribution models considering the impact of LST on turbulence fluxes should use more auxiliary meteorological data to calculate the aerodynamic or surface resistance. These meteorological variables such as air temperature and atmosphere specific humidity lack products with high spatial resolution;[24,57] (2) previous studies show no significant difference among results of these attribution methods, that is, the relative dominance of radiative and non-radiative processes[58,59].

All biophysical sensitivity terms, including albedo ($\frac{d\alpha_{bio}}{dLAI}$), latent heat ($\frac{dLE_{bio}}{dLAI}$), sensible heat ($\frac{dH_{bio}}{dLAI}$), downward shortwave radiation ($\frac{dSW\downarrow_{bio}}{dLAI}$) and downward longwave radiation ($\frac{dLW\downarrow_{bio}}{dLAI}$), are computed at the monthly scale by the same method as LST sensitivity but using 0.1° spatial resolution input data (to match ERA radiation and turbulent fluxes data). Correspondingly, other input data, including the LAI, DEM, and land cover data, are also spatially aggregated to 0.1° resolution. To be consistent with the LST sensitivity, the window size is still set to 50 × 50 km for the 0.1° data input, which is approximately 5 × 5 pixels at the equator.

Moreover, it is worth mentioning that $\frac{dSW\downarrow_{bio}}{dLAI}$ and $\frac{dLW\downarrow_{bio}}{dLAI}$ here are not induced by large-scale atmospheric circulation or biochemical processes but by local or mesoscale biophysical atmospheric feedbacks[60]. Theoretically, greening could also affect non-local climate, through horizontal heat or vapor transfer, and could also induce global cooling by absorbing the atmospheric $CO_2$. However, the radiation sensitivities from our method do not contain the contributions from these two processes, mainly because the spatial regression cannot extract the teleconnected or global uniform signal. In this study, the radiation sensitivity indicates the vertical impact of greening on climate. Specifically, greening directly alters energy absorption as well as water and heat exchange efficiency between the surface and atmosphere, subsequently affecting near-surface temperature, relative humidity, atmospheric transmittance, and cloud cover locally[61], which could further alter surface downward shortwave and longwave radiation.

### Generation of climatological sensitivities

Through the mentioned spatial regression method, we obtain the monthly biophysical sensitivity of LST to LAI from 2001 to 2018 with 0.05° spatial resolution and decomposed sensitivity of energy balance terms to LAI from 2001 to 2015 with 0.1° spatial resolution. To represent the mean condition of the study period, we calculate the mean value for each month over 18 (or 15) years. Before the temporal aggregation, we remove the value within the maximum and the minimum 1% value based on the cumulative distribution frequency (CDF) for all the mean monthly results to exclude the impact from outliers. Then, annual sensitivities are averaged by the monthly value if all monthly sensitivities are valid for a given location. Notably, these climatology sensitivities are calculated for the spatial and seasonal patterns and the decomposition analysis, as they only represent the temperature response to greening under the multi-year average background climate.

### Quantification of the temperature effect of vegetation greening

The LST trend induced by greening ($\delta LST_{bio}$) over the study period (2001–2018) is the result of both biophysical sensitivity and LAI variation. Since the interannual variation of vegetation and climatic

conditions can significantly affect the biophysical sensitivity, we use the original monthly sensitivity rather than the climatological value to estimate the temperature effect of greening. Specifically, all the sensitivity maps are firstly aggregated into 0.5° resolution to eliminate the impact of land cover or land management changes on LAI or sensitivity at the fine scale. Here, only grids with more than 50% valid values are retained. The Harmonic Analysis of Time Series (HANTS) is conducted to fill the invalid values of the sensitivity series for each 0.5° grid[62]. Then, the greening-induced LST variation ($\Delta LST_{bio}$) can be calculated based on year-to-year LAI variation ($\Delta LAI$) and the LST sensitivity ($\frac{dLST_{bio}}{dLAI}$) at monthly scale:

$$\Delta LST_{bio}(y,m) = \Delta LST_{bio}(y-1,m) + \frac{\frac{dLST_{bio}}{dLAI}(y,m) + \frac{dLST_{bio}}{dLAI}(y-1,m)}{2} \times \Delta LAI(y,m)$$

$$(13)$$

$$\Delta LAI(y,m) = LAI(y,m) - LAI(y-1,m) \qquad (14)$$

where $y$ and $m$ denote the year and month. Monthly $\delta LST_{bio}$ could be computed by the linear trend of 17 years $\Delta LST_{bio}$, and the annual or seasonal $\delta LST_{bio}$ are similarly regressed from 17 years annual or seasonal $\Delta LST_{bio}$ (temporally aggregated from the monthly $\Delta LST_{bio}$). The uncertainty of $\delta LST_{bio}$ could be further estimated using the 95% confidence interval of the regression slope of year-to-year greening-induced temperature variation.

In addition, the temperature trend ($\delta T_{obs}$) is also calculated by linear regression to evaluate the climate mitigation effect of biophysical feedback of greening. Notably, to robustly reflect the global warming signal, the time interval for solving $\delta T_{obs}$ is extended by ten years compared to the interval of the assessment of the biophysical effects. Specifically, we use Climate Research Unit (CRU, TS4.04) surface air temperature product from 1991 to 2018 to calculate $\delta T_{obs}$ (Supplementary Fig. 9), because there is no available satellite monthly LST product before 2001 and trends of surface air temperature and LST are very similar on global and regional scales[63]. Here, $\delta T_{obs}$ represent the temperature response to all forcings, which is dominated by the increasing $CO_2$ concentration induced warming, and we use $\delta T_{obs} - \delta LST_{bio}$ to represent the temperature trend without biophysical feedback of greening. Then, the mitigation percentage can be represented by the ratio of greening induced temperature trend to the warming rate without the mitigation of greening $\left(-\frac{\delta LST_{bio}}{\delta T_{obs} - \delta LST_{bio}} \times 100\%\right)$. Furthermore, the uncertainty of this ratio is calculated based on the error propagation of $\delta T_{obs}$ and $\delta LST_{bio}$.

### Köppen–Geiger climate zones

The recently released Köppen-Geiger World map is used to define the climate zones in our analysis, which is based on the temperature and precipitation data from CRU datasets[64]. We aggregate the 31 climate zones into seven major zones (i.e., humid equatorial, dry equatorial, arid, humid temperate, dry temperate, humid cold, and dry cold) (Supplementary Fig. 17).

### Data availability

The data that support the findings of this study are openly available. The MODIS land surface temperature, albedo and land cover data can be downloaded from https://ladsweb.modaps.eosdis.nasa.gov/search/. The leaf area index data are available at http://www.glass.umd.edu/LAI/MODIS/0.05D/ (GLASS), https://zenodo.org/record/4700264 (GLOBMAP), and https://sites.bu.edu/cliveg/datacodes (GIMMS3g). The ERA5-Land reanalysis products are available at https://cds.climate.copernicus.eu/cdsapp#!/dataset/reanalysis-era5-land-monthly-means. The SRTM DEM is available on the Google Earth Engine at https://

developers.google.com/earth-engine/datasets/catalog/CGIAR_
SRTM90_V4. The MODIS snow cover data are from https://nsidc.org/
data/MYD10CM/versions/6. The FluxCom flux data are available fol-
lowing the instructions at https://www.bgc-jena.mpg.de/geodb/
projects/Home.php. The gridded climate data are available at https://
catalogue.ceda.ac.uk/uuid/e0b4e1e56c1c4460b796073a31366980.
The GLDAS 2.1 dataset are available at https://disc.gsfc.nasa.gov/
datasets. The Köppen-Geiger climate classification map is accessible at
http://koeppen-geiger.vu-wien.ac.at/present.htm. Source data are
provided with this paper.

## Code availability
The code used to process and analyze the data is written with Python
3.8 and MATLAB 2019a. The scripts are available at https://zenodo.org/
record/7446790 or from the corresponding author upon request.

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

## Acknowledgements
This work is supported by the National Natural Science Foundation of China (Grant No. 41921001, Z.L., P.Y., and W.W.), the Strategic Priority Research Program of Chinese Academy of Sciences (Grant No. XDA28050200, H.W.) and the National Natural Science Foundation of China (Grant No. 41871267, H.W.).

## Author contributions
Z.L. and Y.L. conceived and designed the research. X.L. and P.L. organized and processed the data. Y.L. and H.W. carried out the analysis, with help from C.Z., P.Y., and W.W. in interpretation of the results. Y.L. drafted the paper. Z.L., C.Z., R.T., G.S., and L.M. edited and revised the manuscript.

## Competing interests
The authors declare no competing interests.
