## [Peer Review File · Nature Communications]

Biophysical Impacts of Earth Greening Can Substantially Mitigate Regional Land Surface Temperature WarmingREVIEWER COMMENTS

Reviewer #1 (Remarks to the Author):

This manuscript presents a comprehensive assessment regarding the climate mitigation of the persistent earth greening in the recent 20 years. The authors first estimate the biophysical sensitivity of temperature to vegetation greenness ($dLST_{bio}/dLAI$) in the stable vegetated area. The key datasets include a statistical-based monthly land surface temperature (LST) and GLASS leaf area index (LAI). The $dLST_{bio}/dLAI$ is then decomposed into the contribution from "radiative" and "non-radiative" processes. This process is based on the energy balance equation and multi-source energy budgets products. Authors then apply the LST sensitivity and the long-term LAI trend to quantify the temperature trend induced by the earth greening (δLST_{bio}). The spatial and seasonal variability of $dLST_{bio}/dLAI$ and δLST_{bio} are exhaustively analyzed and the authors suggest a considerable climate mitigation potential of greening in China and India due to the strong cooling effect.

Overall, this study has a good quality with substantial work behind. The manuscript is well organized and the writing is compelling. I believe it is within the scope of Nature Communications and on the general interest of the readers working in areas of remote sensing modeling and global change. Despite these, I have several comments and suggestions on this article:

(1) LST is an important variable for the analysis of this paper, however, few details are provided for the LST dataset used in this study. It seems the LST products from MODIS Aqua and MODIS Terra satellites are both used for generating the daily LST first. But what is the principle for the daily LST aggregated to the monthly scale? Indeed, the spatial coverage of LST data is significantly improved (Figure S11), but how is the accuracy or consistency with the original MODIS LST product? In addition, it should be mentioned what collection of MODIS LST is used as the input of your "two-step method". As the collection C6 MODIS LST makes several refinements to deal with the sensor degradation issues, the C6 products are expected to be the better choice.

(2) LAI is the other important variable for estimating sensitivity. My major concern of the robustness of the results to the choice of LAI products. The results from different LAI products (such as GLOBMAP, and GIMMS) should be compared to show the findings are robust across multiple LAI products, or at least the authors should illustrate the reason for choosing only GLASS LAI as the model input.

(3) In section 5.2, the window size for estimating the $dLST_{bio}/dLAI$ is set to 50×50 km. The choice of window size seems to be subjective. It should be declared to what extent this model parameter affects your results.

(4) The attribution model in section 5.3 is based on the simple first-order expansion of Stefan Boltzmann equation. However, the major issue of this model is the neglect of the relationship between LST and turbulent fluxes. For instance, the sensible heat $(T_s - T_a)/\rho C_p$ is expressed as the function of T_s (LST). Thus, how does this issue of the decomposing model affect the final attributing results?

(5) In this paper, the authors only calculate the sensitivity for the "stable" vegetated area. In other words, the LAI variation induced by land cover changes is ignored in this study. However, in section 3.4, the authors state that the increased harvested area and afforestation project lead to the cooling effect in India and China. Should not the increased harvested area and afforestation project be treated as two examples of land cover change?

(6) The authors compared the δLST_{bio} and the observed LST trend over the same period to quantify the climate mitigation effect of greening. However, the LST trend in figure s9 seems contradictory to the common knowledge of global warming. For instance, a large

area with cooling signal is found in North America, leading to the negative LST trend in Canada in figure 6. The author should consider a longer time interval for calculating the LST trend to further reduce the uncertainty of the ratio in figure 6.

(7) In section 3.2, the $dLST_{bio}/dLAI$ is decomposed into non-radiative, radiative, and indirect climatic feedbacks. However, only the non-radiative and radiative feedbacks are discussed in the following parts. What about the indirect climatic feedback? From figure 3, I can see that this part has a very low impact, but additional descriptive sentences are needed in the article.

(8) As shown in figure 1b, the $dLST_{bio}/dLAI$ has a good relation with background climate (temperature and precipitation). Why did the authors choose snow cover, LAI, and radiation as the indicators in the attribution analysis (section 3.2), but not discuss the impact of temperature and precipitation on $dLST_{bio}/dLAI$? In addition, in figure 4a-c, why the mean sensitivity from January to April is used to plot the scatters, but not the annual value in figure 4d-f?

(9) In line 273, the warming trend induced by forest greening is 0.001 K/decade. The number seems wrong compared with the plot in figure 5.

Reviewer #2 (Remarks to the Author):

I'm sorry to say that I find little new in this manuscript. We already know these results (Refs. 4, 9, 10, 11, 14, 17, etc.). This manuscript is just repeating.

Their results (L68-87) were very similar compared to Ref. 9 (Fig. 1a vs. Fig. 3D of Ref. 9). A similar regression analysis was used, which I doubt unveils complex two-way effects between vegetation growth and climate change. I did not see many mechanisms in the methodology and discussion. For example, how do the authors resolve the following causal ambiguity (as noted in Ref. 20): an increase in T_{air} can increase LST. An increase in T_{air} can increase LAI in colder regions but may lead to a decrease in LAI in hotter regions. The analysis in this paper can only be attributed to a change in LAI, but this is indeed due to a change in T_{air} . These issues are arguably critical in Figure 2b.

The content of L98-118 is also similar to ref. 10. For example, Figure 1c with Table 1 of ref. 10.

In addition, talking about these sensitivities requires a stable background climate and vegetation growth (or very small changes). If there are significant changes in the background climate and vegetation growth, these sensitivities should also change. Therefore, the analysis in Section 3.3 is not convincing. A simple example is given to illustrate my concern.

Let $y = x^4 + z$, and assume z is independent of x .

Then, $dy/dx = 4x^3$;

When $x = 1$, $dy/dx = 4$;

When $x = 2$, $dy/dx = 32$;

The sensitivity of LST to LAI is similar to dy/dx , which changes if x is changing. You can only assume the sensitivity is a constant if x does not change much. Here y is analog to LST, and x is analog to LAI (x could also be the background climate).

Comments on the methodology:

There may be several problems with the methods section.

First, I doubt that the statistical approach is able to decompose the multiple treatments involved in the surface energy balance, as I elaborated above.

I did not see a detailed description of the linear regression model.

L55: How exactly does the spatially moving widow strategy work? It's too vague, more description is needed.

I can't understand equations 2 and 3. They are not the standard equations for LE and H. How do you ensure that the turbulent flux changes in your analysis are caused by LAI changes and not by the energy closure preprocessing of these two equations?

Equations 6-10, why are total derivatives with respect to LAI? For example, there may be a variety of factors that affect changes in LE and H, such as wind speed, potential air temperature, changes in incident shortwave radiation, precipitation, soil moisture, VPD, etc. Assuming that LAI is the only driver of these surface changes is clearly invalid.

L589-590: Why is it not induced by large-scale atmospheric circulation and biochemical processes? I cannot understand.

Investigating seasonal energy balance should not ignore surface heat fluxes. They may have a significant contribution, especially in colder regions.

Comments specific to the line.

L155: Which two processes?

L170: Why are all changes in snow attributed to changes in LAI?

Reviewer #3 (Remarks to the Author):

Review of "Biophysical impacts of Earth greening can mitigate up to 50% of the land surface temperature warming"

This study applies an observationally based framework to quantify the temperature effects, globally and regionally, of the observed global greening trend over the past 20 years driven by biophysical mechanisms. Furthermore, the sensitivity of the temperature response is decomposed into contributions from radiative, non-radiative, and indirect climate feedbacks to delineate the underlying driving factors, as well as differences between vegetation types. The study generally confirms the main features of biophysical effects of vegetation changes found in previous work but adds value by using a purely observation-based approach and applying it to a less well studied case of greening rather than large-scale land use conversions. Overall, the paper is well structured with good figures. I find the study interesting, providing details on biome and process level that gives it the potential to be of importance for understanding the responses in further modeling-based work. Below are a few comments and questions for the authors' consideration before I can recommend publication, in particular related to treatment of uncertainties in the numbers and how the results are formulated in the context of mitigation.

General comments:

- There is no uncertainty/error estimate in any of the numbers provided, whereas Fig. 1 show substantial standard errors, in some cases even in terms of sign sensitivity with vegetation type. This should preferably be quantified and at the very least better discussed, including in the abstract.

- It is not immediately clear to me if the percentage numbers given are the fraction of the total temperature impact due to greening only that is "mitigated" or the fraction of the total observed warming (i.e. total human-induced). Should be clearly written in the text throughout.

- The title gives impression of a very large mitigation potential of the greening. Upon seeing the findings, I feel it should be explicit about this being local/regional effects (and depending on the answer to the comment above, maybe further revised). Later parts of the paper states that the global magnitude of biophysically based mitigation is limited.
- The results are generally discussed in terms of mitigation and the authors for instance say that the cooling effect derived over India and China can mitigate up to 50% of global warming. But how confident are the authors in the potential for extrapolating these results based on historically observed changes? I.e. does dynamic vegetation changes or saturation effects come into play? Strictly, one would only be able to state that these changes have offset some of the total warming, which maybe should be reflected in the wording.
- Related: from previous studies, it is my impression that the observed global greening is primarily driven by fertilization effects due to increased CO₂, which makes it more a consequence of our emissions. How actively can we continue to enhance the greening without exacerbating the CO₂ or other pollution? Should perhaps a mention of the driving factors of greening be included? Some discussion of irrigation and forestation is there, but these are the types of conversions that the authors explicitly say they do not address here...
- Is LAI a sufficient variable? How about SAI – would that factor into the, in particular, albedo changes? Moreover, there are a number of satellite-derived products out there, with large discrepancies suggested by some. A word on the GLASS LAI compared to other products would be useful.

Specific:

Line 36: LUC – should it rather be the more commonly used LULCC?

Line 64: this could be a good place to clarify what temperature signals are compared, i.e. is this the total observed temperature change relative to some baseline climatology or the total temperature signal due to greening alone?

Line 87: seems like a significant difference - can the authors offer some suggestions for why?

Line 243: Could be useful with a figure that show the regional ratio of biophysically-driven to total (as in the current Fig. S9) temperature

Line 245: does this also affect the total temperature impact of the greening and hence the ratio/fraction of warming that is offset?

Throughout: some typos that need fixing...

Response to Reviewers' Comments

We greatly appreciate the constructive comments from the anonymous reviewers. They have made valuable comments and suggestions, according to which the manuscript has been revised and improved. Below are the point-by-point responses to the comments, along with the revision of the manuscript (typed in italics) and the location of the revision (typed in bold). Hope the revision will make it more acceptable for publication.

Reviewer #1 (Remarks to the Author):

This manuscript presents a comprehensive assessment regarding the climate mitigation of the persistent earth greening in the recent 20 years. The authors first estimate the biophysical sensitivity of temperature to vegetation greenness ($dLST_{bio}/dLAI$) in the stable vegetated area. The key datasets include a statistical-based monthly land surface temperature (LST) and GLASS leaf area index (LAI). The $dLST_{bio}/dLAI$ is then decomposed into the contribution from “radiative” and “non-radiative” processes. This process is based on the energy balance equation and multi-source energy budgets products. Authors then apply the LST sensitivity and the long-term LAI trend to quantify the temperature trend induced by the earth greening (δLST_{bio}). The spatial and seasonal variability of $dLST_{bio}/dLAI$ and δLST_{bio} are exhaustively analyzed and the authors suggest a considerable climate mitigation potential of greening in China and India due to the strong cooling effect. Overall, this study has a good quality with substantial work behind. The manuscript is well organized and the writing is compelling. I believe it is within the scope of Nature Communications and on the general interest of the readers working in areas of remote sensing modeling and global change.

Response: We appreciate all the positive comments by reviewer #1. We have carefully considered all the comments and suggestions and made corresponding point-by-point responses.

Despite these, I have several comments and suggestions on this article:

(1) LST is an important variable for the analysis of this paper, however, few details are provided for the LST dataset used in this study. It seems the LST products from MODIS Aqua and MODIS Terra satellites are both used for generating the daily LST first. But what is the principle for the daily LST aggregated to the monthly scale? Indeed, the spatial coverage of LST data is significantly improved (Figure S11), but how is the accuracy or consistency with the original MODIS LST product? In addition, it should be mentioned what collection of MODIS LST is used as the input of your “two-step method”. As the collection C6 MODIS LST makes several refinements to deal with the sensor degradation issues, the C6 products are expected to be the better choice.

Response: Thank you for raising these questions. The principle for “daily LST aggregated to the monthly scale” can be found in Ding et al (2020), which has been added to the reference in the revised manuscript. For the data collection, we would like to clarify that the products are from collection 6. To clearly show how the monthly LST is generated, we have made a new figure show the overall production flow of the monthly LST (Supplementary Fig. 11), and the revised description of monthly LST data production is as follows (Lines 451–462):

The monthly mean land surface temperature (LST) over the study period (2001–2018) is generated through a two-step strategy. The original LST data are from the collection 6 MOD11A1 and MYD11A1 products with a 1-km spatial resolution downloaded from the National Aeronautics and Space Administration (NASA) website. Based on the quality control flag, only data with LST errors less than 2 K are used. First, the instantaneous temperature observations are converted to the daily mean temperature using a statistical-based method (Xing et al., 2021). This method generates the daily mean LST for each pixel using the weighted mean value of different MODIS observation combinations (at least one daytime and one nighttime

observation for both Aqua and Terra observations). Second, the daily average temperature is temporally aggregated to the monthly value using averaged by day (ABD) approach (Ding et al., 2020) and spatially aggregated to 0.05° resolution. The flow chart for the monthly LST production can be found in Supplementary Fig. 11.

Supplementary Figure 11. Flow chart for the monthly LST production.

We also compared our monthly LST data with the mean of daytime and nighttime LST values from MYD11C3 (MODIS/Aqua Land-Surface Temperature/Emissivity Monthly Global 0.05Deg CMG). This averaging method for MYD11C3 products has been used for related studies (Alkama and Cescatti, 2016; Duveiller et al., 2018). Take the data of 2010 as an example, we find good consistency between the two monthly datasets with the R^2 value larger than 0.99 for all months (see Table. R1), although the original MODIS LSTs are derived from different algorithms (split-window method for MOD11A1 and MYD11A1, and physics-based day/night method for MYD11C3). However, our monthly LSTs are systematically lower than averaged MYD11C3 LSTs (negative bias could be found in Table. R1). The main reason is the parameter in Xing et al. (2021) is optimized using in-situ daily mean temperature samples, which are calculated from all weather conditions. However, the mean value of MODIS Aqua daytime LST and nighttime LST represent the daily mean LST for the theoretical clear sky. This difference is systematical and should not have an impact on our results, as we use the spatial relative difference to estimate the sensitivity.

	Jan	Feb	Mar	Apr	May	Jun	Jul	Aug	Sep	Oct	Nov	Dec
R^2	0.997	0.997	0.997	0.998	0.998	0.998	0.999	0.998	0.998	0.998	0.995	0.997
Bias (K)	-2.22	-2.22	-2.09	-1.76	-1.63	-1.73	-1.70	-1.84	-1.80	-1.84	-1.94	-2.10

Table. R1. Comparison of our monthly LST and averaged LST from MYD11C2

Reference:

Alkama, R., Cescatti, A., 2016. Biophysical climate impacts of recent changes in global forest cover. *Science* (80-.). 351, 600–604. <https://doi.org/10.1126/science.aac8083>

Ding, F., Savtchenko, A., Hearty, T., Wei, J., Theobald, M., Vollmer, B., Tian, B., Fetzer, E., 2020. Assessing the impacts of two averaging methods on air level 3 monthly products and multiyear monthly means. *J. Atmos. Ocean. Technol.* 37, 1027–1050. <https://doi.org/10.1175/JTECH-D-19-0129.1>

Duveiller, G., Hooker, J., Cescatti, A., 2018. The mark of vegetation change on Earth's surface energy balance. Nat. Commun. 9, 64–75. <https://doi.org/10.1038/s41467-017-02810-8>

Xing, Z., Li, Z.L., Duan, S.B., Liu, X., Zheng, X., Leng, P., Gao, M., Zhang, X., Shang, G., 2021. Estimation of daily mean land surface temperature at global scale using pairs of daytime and nighttime MODIS instantaneous observations. ISPRS J. Photogramm. Remote Sens. 178, 51–67. <https://doi.org/10.1016/j.isprs.2021.05.017>

(2) LAI is the other important variable for estimating sensitivity. My major concern of the robustness of the results to the choice of LAI products. The results from different LAI products (such as GLOBMAP, and GIMMS) should be compared to show the findings are robust across multiple LAI products, or at least the authors should illustrate the reason for choosing only GLASS LAI as the model input.

Response: Thank you for the constructive comment. Here, we use GLASS LAI as the main LAI product for analysis for the following reasons:

- (I) GLASS LAI is based on consistent MODIS reflectance observations and has been proved to have high accuracy (Xiao et al., 2017).
- (II) GLASS LAI is a spatiotemporally continuous product, which is important for the calculation of temperature sensitivity.
- (III) The spatial resolution of GLASS LAI can match our monthly temperature product.

We also performed additional sensitivity tests for different LAI inputs (GLASS, GLOBMAP, GIMMS), which shows the robustness of our results. The description of sensitivity tests is as follows (**Supplementary Discussion 3**):

We performed sensitivity tests for the choice of LAI datasets and the size of moving window. Specifically, we calculated the monthly sensitivity using additional GIMMS and GLOBMAP (resampled into 0.05°) LAI datasets. Here, we did not test the MODIS C6 product, due to the missing data issue in high-latitude winter for the main look-up-table method. We compared the final temperature effect but not LST sensitivity, because differences can be found among the long-term trends from different LAI products (Jiang et al., 2017). We find almost the same spatial pattern and good latitudinal consistency of greening induced temperature effect (δLST_{bio}) from the three products (Supplementary Fig.13, 14) at seasonal or annual scale.

Supplementary Figure 13. Spatial maps of greening induced LST trend (δLST_{bio}) estimated by GLASS, GIMMS and GLOBMAP LAI datasets. a Annual mean, b JJA (June to August) mean and c DJF (December to February) mean of δLST_{bio} from GLASS. d to f, same as a to c, but for GIMSS δLST_{bio} . g to i, same as a to c, but for GLOBMAP δLST_{bio} .

Supplementary Figure 14. **Latitudinal patterns of greening induced LST trend (δLST_{bio}) estimated by GLASS, GIMMS and GLOBMAP LAI datasets.** Latitudinal patterns of **a** annual mean, **b** JJA (June to August) mean, and **c** DJF (December to February) mean of greening-induced LST trend (δLST_{bio}) estimated by three LAI products.

Reference:

Xiao, Z., Liang, S., Jiang, B., 2017. Evaluation of four long time-series global leaf area index products. *Agric. For. Meteorol.* 246, 218–230. <https://doi.org/10.1016/j.agrformet.2017.06.016>

(3) In section 5.2, the window size for estimating the $dLST_{bio}/dLAI$ is set to 50×50 km. The choice of window size seems to be subjective. It should be declared to what extent this model parameter affects your results.

Response: Thanks for your valuable comment. The selection of window size does not significantly affect our results. We performed sensitivity tests for different sizes, the description is as follows (**Supplementary Discussion 3**):

Similarly, we repeated experiments for different window sizes of 40 km, 50 km and 60 km (about 7×7 , 9×9 and 11×11 pixels near the equator, respectively) using only GLASS LAI. Reduced window size means fewer samples for regression and thus may generate higher uncertainty. However, the LST sensitivity was not significantly affected by the changing window size. We find almost the same latitudinal pattern, with all the scatters near the 1:1 line. (Supplementary Fig.15). These results show our results are robust against different LAI products and window sizes.

9

Supplementary Figure 15. Comparison of LST sensitivity with different window sizes (40 km, 50 km, 60 km). a to c Latitudinal patterns of LST sensitivity for annual mean, JJA (June to August) mean, and DJF (December to February) mean, respectively. d to f Density scatter plots between LST sensitivities derived from 50 km and 40 km for annual mean, JJA mean and DJF mean, respectively, respectively. g to i Same as d to f, but for the comparison between window sizes of 50 km and 60 km.

(4) The attribution model in section 5.3 is based on the simple first-order expansion of Stefan Boltzmann equation. However, the major issue of this model is the neglect of the relationship between LST and turbulent fluxes. For instance, the sensible heat $(T_s - T_a)/\rho C_p$ is expressed as the function of T_s (LST). Thus, how does this issue of the decomposing model affect the final attributing results?

Response: Thank you for pointing this out. Neglecting the relationship between LST and turbulent fluxes does not affect the attribution result. We have added an explanation of this issue in this section as follows (Lines 610–617):

Meanwhile, we neglect the complex effect of LST on other energy balance terms in the attribution model for the following two reasons: (1) attribution models considering the impact of LST on turbulence fluxes should use more auxiliary meteorological data to calculate the aerodynamic or surface resistance. These meteorological variables such as air temperature and atmosphere specific humidity lack products with high spatial resolution (Lee et al., 2001; Rigden et al., 2017); (2) previous studies show no significant difference among results of these attribution methods, that is, the relative dominance of radiative and non-radiative processes (Wang et al., 2018; Wang et al., 2020).

Reference:

Lee, X., Goulden, M.L., Hollinger, D.Y., Barr, A., Black, T.A., Bohrer, G., Bracho, R., Drake, B., Goldstein, A., Gu, L., Katul, G., Kolb, T., Law, B.E., Margolis, H., Meyers, T., Monson, R., Munger, W., Oren, R., Paw U, K.T., Richardson, A.D., Schmid, H.P., Staebler, R., Wofsy, S., Zhao, L., 2011.

Observed increase in local cooling effect of deforestation at higher latitudes. *Nature* 479, 384–387. <https://doi.org/10.1038/nature10588>

Rigden, A.J., Li, D., 2017. Attribution of surface temperature anomalies induced by land use and land cover changes. *Geophys. Res. Lett.* 44, 6814–6822. <https://doi.org/10.1002/2017GL073811>

Wang, L., Lee, X., Schultz, N., Chen, S., Wei, Z., Fu, C., Gao, Y., Yang, Y., Lin, G., 2018. Response of Surface Temperature to Afforestation in the Kubuqi Desert, Inner Mongolia. *J. Geophys. Res. Atmos.* 123, 948–964. <https://doi.org/10.1002/2017JD027522>

Wang, L., Tian, F., Wang, X., Yang, Y., Wei, Z., 2020. Attribution of the land surface temperature response to land-use conversions from bare land. *Glob. Planet. Change* 193, 103268. <https://doi.org/10.1016/j.gloplacha.2020.103268>

(5) In this paper, the authors only calculate the sensitivity for the “stable” vegetated area. In other words, the LAI variation induced by land cover changes is ignored in this study. However, in section 3.4, the authors state that the increased harvested area and afforestation project lead to the cooling effect in India and China. Should not the increased harvested area and afforestation project be treated as two examples of land cover change?

Response: Thank you for raising this question. We quite agree that land use and land cover change (LULCC) is an important driver of global vegetation greening. In the revised manuscript, the temperature trend results related to vegetation greening (δLST_{bio}) are from all vegetated pixels, rather than “stable” pixels. Specifically, additional spatial aggregation is adopted to deal with the potential abrupt changes of sensitivity due to land cover change. The description is as follows (**Line 658–665**):

Since the interannual variation of vegetation and climatic conditions can significantly affect the biophysical sensitivity, we used the original monthly sensitivity rather than the climatological value to estimate the temperature effect of greening. Specifically, all the sensitivity maps are firstly aggregated into 0.5° resolution to eliminate the impact of land cover or land management changes on LAI or sensitivity at the fine scale. Here, only grids with more than 50% valid values are retained. The Harmonic Analysis of Time Series (HANTS) was conducted to fill the invalid values of the sensitivity series for each 0.5° grid (Jiang et al., 2017).

Reference:

Jiang, C., Ryu, Y., Fang, H., Myneni, R., Claverie, M., Zhu, Z., 2017. Inconsistencies of interannual variability and trends in long-term satellite leaf area index products. *Glob. Chang. Biol.* 23, 4133–4146. <https://doi.org/10.1111/gcb.13787>

(6) The authors compared the δLST_{bio} and the observed LST trend over the same period to quantify the climate mitigation effect of greening. However, the LST trend in figure s9 seems contradictory to the common knowledge of global warming. For instance, a large area with cooling signal is found in North America, leading to the negative LST trend in Canada in figure 6. The author should consider a longer time interval for calculating the LST trend to further reduce the uncertainty of the ratio in figure 6.

Response: Thank you for raising these important points. Unfortunately, our assessment cannot be extended to a longer period, mainly due to the lack of consistent and continuous satellite observation LST products before 2001. Meanwhile, we would like to politely argue that the time period of our study is sufficient to obtain a reliable estimate of the temperature effect of vegetation greening. To address the large uncertainty in the mitigation ratio caused by the uncertainty of temperature trends, we made the comparison using Climate Research Unit (CRU) temperature trend from 1991 to 2018 to ensure the global warming trend is robust. We have also provided an assessment of the uncertainty of the ratio. Specific modifications to the method are as follows (**Lines 682–696**):

In addition, the temperature trend (δT_{obs}) is also calculated by linear regression to evaluate the climate

mitigation effect of biophysical feedback of greening. Notably, to robustly reflect the global warming signal, the time interval for solving δT_{obs} is extended by ten years compared to the interval of the assessment of the biophysical effects. Specifically, we use Climate Research Unit (CRU) surface air temperature product from 1991 to 2018 to calculate δT_{obs} (Supplementary Fig. 9), because there is no available satellite monthly LST product before 2001 and trends of surface air temperature and LST are very similar on global and regional scales (Wang et al., 2022). Here, δT_{obs} represent the temperature response to all forcings, which is dominated by the increasing CO_2 concentration induced warming, and we use $\delta T_{obs} - \delta LST_{bio}$ to represent the temperature trend without biophysical feedback of greening. Then, the mitigation percentage can be represented by the ratio of greening induced temperature trend to the warming rate without the mitigation of greening $\left(-\frac{\delta LST_{bio}}{\delta T_{obs} - \delta LST_{bio}} \times 100\%\right)$. Meanwhile, the uncertainty of this ratio is further calculated based on the error propagation of δT_{obs} and δLST_{bio} .

(7) In section 3.2, the $dLST_{bio}/dLAI$ is decomposed into non-radiative, radiative, and indirect climatic feedbacks. However, only the non-radiative and radiative feedbacks are discussed in the following parts. What about the indirect climatic feedback? From figure 3, I can see that this part has a very low impact, but additional descriptive sentences are needed in the article.

Response: Thank you for your suggestion. We have added a descriptive sentence for the indirect climatic feedback as follows (Lines 162–164):

As shown in Fig. 3a, we find a minor impact from indirect climatic feedback (green line) for all latitudes, suggesting the local temperature effect of vegetation greening is mainly induced by the direct modification of surface biophysical parameters.

(8) As shown in figure 1b, the $dLST_{bio}/dLAI$ has a good relation with background climate (temperature and precipitation). Why did the authors choose snow cover, LAI, and radiation as the indicators in the attribution analysis (section 3.2), but not discuss the impact of temperature and precipitation on $dLST_{bio}/dLAI$? In addition, in figure 4a-c, why the mean sensitivity from January to April is used to plot the scatters, but not the annual value in figure 4d-f?

Response: Thank you for your comment. Temperature and precipitation show good relationships with LST sensitivity do not mean their changes are the direct causes of biophysical sensitivity. This is the reason why we need a decomposition analysis of the temperature sensitivity, as we write in the revised manuscript as follows (Lines 153–154):

The LST sensitivity varies significantly with vegetation type and climatic conditions, but the biophysical mechanism behind is still unclear.

For figure 4a-c, mean sensitivity from January to April is used because this snow-related albedo warming only occurs in such months, as we mentioned in line 165.

In addition, we give more explanation of how LAI change directly affects biophysical parameters and how it further affects temperature. Theoretically, vegetation greening directly affects surface albedo, aerodynamic resistance, and surface resistance ($\partial x/\partial LAI$) to modify the energy absorption, distribution, and diffusion processes, then cause the local temperature variation ($\partial LST/\partial x$).

For the albedo pathway, the partial derivatives of LST to albedo ($\partial LST/\partial \alpha$) is obviously linearly related to

the downward shortwave radiation, as the albedo is the percentage of reflected energy. Greening affects albedo ($\partial\alpha/\partial LAI$) through covering more background surface area (usually higher albedo soil) with the vegetation canopy (lower albedo). According to the Beer-Lambert Law, the canopy transmittance (thereby canopy albedo) is driven by LAI (Ni-Meister et al., 2010):

$$\partial\alpha/\partial LAI \sim -\exp(-kLAI)$$

where k is a parameter related to canopy structure such as leaf orientation distribution. In addition, surface albedo under the canopy is also important (Li et al., 2017). Specifically, $\partial\alpha/\partial LAI$ become extremely sensitive if the background is snow. Hence, we argue that greening-induced albedo change in snow regions can dominate the positive temperature sensitivity.

For the resistances, greening affect aerodynamic resistance (R_a) mainly by altering the roughness lengths and displacement height, and this sensitivity ($\partial R_a/\partial LAI$) is nonlinearly correlated to the wind speed and LAI:

$$\partial R_a/\partial LAI \sim f(LAI)/u$$

LAI also modifies the canopy conductance of vegetation and then the surface resistance (R_s) of the hybrid pixel. In fact, the early model and FAO-56 characterize R_s in Penman-Monteith equation by a simple linear function of LAI (Cleugh et al., 2007), which assumes a constant $\partial R_s/\partial LAI$ value. The R_s model is further refined through a two-source framework and a better representation of canopy conductance for vegetation components (Leuning et al., 2008), leading to the dependence of $\partial R_s/\partial LAI$ on both LAI and vapor pressure deficit (VPD):

$$\partial R_s/\partial LAI \sim f(LAI)f(VPD)$$

As for the partial derivatives of LST to resistances ($\partial LST/\partial R_a$ and $\partial LST/\partial R_s$), its complex expressions can refer to the supplementary information of A. J. Rigden and Li (2017). Here, we use fluxnet data to show the potential relationship between the sensitivity and drivers (Fig. R2). We find a linear relationship between $\partial LST/\partial R_a$ and shortwave radiation (SW) and an inverse proportional relationship between R_s and $\partial LST/\partial R_s$. However, no significant relationship is found between other climatic drivers and LST sensitivity to R_a or R_s .

Figure. R2. Scatter plots between LST sensitivities (to R_a and R_s) and potential drivers

Based on the facts above, we argue that LAI (affecting $\partial\alpha/\partial LAI$, $\partial R_a/\partial LAI$, $\partial R_s/\partial LAI$ and $\partial LST/\partial R_s$), shortwave radiation (affecting $\partial LST/\partial\alpha$ and $\partial LST/\partial R_a$), and snow cover (affecting $\partial\alpha/\partial LAI$) are the drivers of LST sensitivity. We admit that air temperature and precipitation are the fundamental reason for LAI and snow cover changes, but they only show the indirect impact on the sensitivity.

Reference:

- Cleugh, H.A., Leuning, R., Mu, Q., Running, S.W., 2007. Regional evaporation estimates from flux tower and MODIS satellite data. *Remote Sens. Environ.* 106, 285–304. <https://doi.org/10.1016/j.rse.2006.07.007>
- Leuning, R., Zhang, Y.Q., Rajaud, A., Cleugh, H., Tu, K., 2008. A simple surface conductance model to estimate regional evaporation using MODIS leaf area index and the Penman-Monteith equation. *Water Resour. Res.* 44. <https://doi.org/10.1029/2007WR006562>
- Li, Q., Ma, M., Wu, X., Yang, H., 2017. Snow Cover and Vegetation-Induced Decrease in Global Albedo From 2002 to 2016. *J. Geophys. Res. Atmos.* 124–138. <https://doi.org/10.1002/2017JD027010>
- Ni-Meister, W., Yang, W., Kiang, N.Y., 2010. A clumped-foliage canopy radiative transfer model for a global dynamic terrestrial ecosystem model. I: Theory. *Agric. For. Meteorol.* 150, 881–894. <https://doi.org/10.1016/j.agrformet.2010.02.009>
- Rigden, A.J., Li, D., 2017. Attribution of surface temperature anomalies induced by land use and land cover changes. *Geophys. Res. Lett.* 44, 6814–6822. <https://doi.org/10.1002/2017GL073811>

(9) In line 273, the warming trend induced by forest greening is 0.001 K/decade. The number seems wrong compared with the plot in figure 5.

Response: Thank you for pointing out this mistake and we feel so sorry for our carelessness. In the revised manuscript, we have used a new method to estimate the greening-induced temperature trend according to the major comments (3) of reviewer #2. We have updated this numbers as follows (**Lines 300–301**):

Conversely, the greening of forests could accelerate global warming by 0.009 ± 0.004 K/decade.

Reviewer #2 (Remarks to the Author):

I'm sorry to say that I find little new in this manuscript. We already know these results (Refs. 4, 9, 10, 11, 14, 17, etc.). This manuscript is just repeating.

Response: We have carefully read the mentioned references, as well as other related research articles. Ref. 4, 8, 14, 23, 24, 31, 47 assessed the biophysical temperature effect of forests, regionally or globally. Ref. 6 and 17 are about the local temperature effect of land use/land cover changes (LULCC). These studies are closely related to earth greening, but greening is a more widespread process involving all vegetation types and areas without LULCC. In comparison, the goal of our study is closer to ref. 9, 10, 11, as you mentioned. However, there are still shortcomings in these studies:

- I) The model-based (ref. 10, 11) results should be carefully treated and verified by observation-based evidence, as model uncertainty may lead to a biased assessment of the net temperature effect of greening. For instance, both these two studies failed to produce the warming effect of greening in boreal regions, possibly because the process of snow masking of vegetation canopies is not well represented in LSMs (Piao et al., 2020).
- II) The methodology of the mentioned observation-based study (ref. 9) has been argued whether it can solve the causality issue between vegetation and temperature.

In general, the existing studies have no consistent conclusions on the temperature of greening at regional or global scale, particularly between the remote sensing-based and model-based studies. Considering these shortcomings, we conducted this study initially to reconcile the disagreement between observation-based and model-based results. To this end, we present a new method to quantify the temperature sensitivity to LAI, which could resolve the causality issue of ref. 9 (please see the response to your next comment). Meanwhile, thanks to the high resolution of remote sensing observations, this is the first time that the temperature sensitivity to greening has been quantified globally at 0.05° resolution. In addition, we provide the first data-driven assessment of the seasonal patterns of the temperature sensitivity globally. By introducing the energy balance equation, we attributed this greening-related temperature effect and revealed the potential driving factors through a physical-based approach. Furthermore, we find a considerable magnitude of greening-induced cooling in China and India, which suggests that human-induced vegetation changes can have strong climate benefits. All these make our study unique and we believe our work is of sufficient novelty, but not simply repeating.

(1) Their results (L68-87) were very similar compared to Ref. 9 (Fig. 1a vs. Fig. 3D of Ref. 9). A similar regression analysis was used, which I doubt unveils complex two-way effects between vegetation growth and climate change. I did not see many mechanisms in the methodology and discussion. For example, how do the authors resolve the following causal ambiguity (as noted in Ref. 20): an increase in Tair can increase LST. An increase in Tair can increase LAI in colder regions but may lead to a decrease in LAI in hotter regions. The analysis in this paper can only be attributed to a change in LAI, but this is indeed due to a change in Tair. These issues are arguably critical in Figure 2b.

Response: Thank you for raising this question. The role of climate change affecting vegetation growth in two-way effects is excluded, mainly because the regression samples of our method are from spatial nearby pixel, and thus share the same background climate. We have emphasized in the revised manuscript (methodology section) that the impact of background climate variation is excluded in our method (**Lines 529–535**):

Similarly, we assume vegetation greenness can be the only driving factor of LST spatial variation under certain restrictions, the biophysical sensitivity of LST to LAI could be regressed from spatial nearby LAI and

LST observations. The advantage of this method over the temporal regression strategy (Forzieri et al., 2017) is the exclusion of the impact of climate natural variability or the long-term warming trend on vegetation growth, because pixels with different LAI values within the moving windows share the same background climate (Supplementary Fig. 16).

Moreover, we made an additional explanation of the mechanisms and a method comparison in **Supplementary Discussion 2 (Sensitivity from different regression methods)**. Here, multi-linear regression (MLR) indicates the method used in ref. 9.

The intrinsic difference between our method and MLR method is the source of regression samples. Specifically, the samples in our studies are from the spatial nearby pixels (sharing a similar background climate), while samples in MLR method are from the time series observations for a given location. The feasibility of temporal regression is debatable mainly because of the changing background climate (Li et al., 2018). The long-term trends and fluctuations of the climate system drive both LAI and LST variation, and thus vegetation and temperature show complex two-way effects (Supplementary Fig. 16a). However, samples from our spatial regression method sharing the same background climate, which means the signal of climate natural variability or the long-term warming trend affecting vegetation growth is excluded. The spatial variability of LST samples is mainly driven by vegetation growth (LAI affecting biophysical properties) after filtering out the impact of land cover and altitude (Supplementary Fig. 16b). Hence, our method is essentially a spatial controlled experiment, but the MLR method is an observation-based statistical method.

Supplementary Figure 16. Schematic representation of the relationship between LAI and LST from the temporal and spatial regression methods. **a** Temporal regression. **b** Spatial regression.

Further comparison of annual sensitivity from our method ($\frac{dLST_{bio}}{dLAI}$) and MLR method ($\frac{\partial LST_{bio}}{\partial LAI}$) can be found in Supplementary Fig. 10. Notably, same LAI and LST datasets were used here. Compared with our result, the annual sensitivity derived from MLR shows a stronger negative signal in the southern mid-latitude, which is mainly caused by the sensitivity difference in Australia. Meanwhile, the positive sensitivity is significantly strengthened in boreal regions in MLR result. These differences are possible because the reversed “temperature-vegetation” effect is superimposed with the “vegetation-temperature” signal of our

concern. For instance, global warming could be the major cause of greening and the stronger boreal signal in MLR is more likely the reflection of temperature influencing the vegetation growth. Conversely, the larger negative sensitivity in Australia is due to the reversed “temperature limits vegetation growth” effect overlapping with the “vegetation greening induces cooling effect”. However, our spatial regression method can exclude the reversed signal of large-scale warming affecting vegetation physiology and phenology, as the sensitivity is regressed from simultaneous LST and LAI observations from spatial samples sharing the same background climate (see Methods).

Supplementary Figure 10. Comparison of annual biophysical sensitivity of LST to LAI derived from multi-linear regression method ($\frac{\partial LST_{bio}}{\partial LAI}$) and our method ($\frac{dLST_{bio}}{dLAI}$). **a** Spatial map of $\frac{\partial LST_{bio}}{\partial LAI}$. **b** Comparison of latitudinal patterns between $\frac{\partial LST_{bio}}{\partial LAI}$ and $\frac{dLST_{bio}}{dLAI}$ (Fig. 1a).

Nevertheless, we fully understand your doubt whether the reversed “temperature→vegetation” effect could still play a major role in terms of space. For instance, in boreal winter, the nearby pixel has higher air temperature, and thus shows a higher LAI value than the center pixel of the moving window (Fig. 2b). We admit that this reverse signal may, to some extent, lead to the overestimation of LST sensitivity in the boreal dormant season. However, considering the weak greening trend in boreal winter, this effect has a very limited role in the final temperature trend (δLST_{bio}) results.

In addition, we present additional evidence to show the positive temperature response to vegetation greening in boreal regions can be induced by the reduction of albedo. Here, we first confirm the annual boreal warming signal is induced by the positive sensitivity from November to May of (Fig. R3).

Figure. R3. Seasonal pattern of greening induced LST trend and LST sensitivity in boreal regions (>50°N)

For such seven months, we compare the total mean LST sensitivity ($\frac{dLST_{bio}}{dLAI}$) with the part of sensitivity contributed by the vegetation-albedo-temperature pathway ($\frac{dLST_{bio}^{\alpha}}{dLAI}$). We find a significant positive

relationship between $\frac{dLST_{bio}}{dLAI}$ and $\frac{dLST_{bio}^{\alpha}}{dLAI}$. Meanwhile, the slope between $\frac{dLST_{bio}}{dLAI}$ and $\frac{dLST_{bio}^{\alpha}}{dLAI}$ is lower than 1 (0.62), which suggests the potential warming induced by albedo feedback is higher than the final warming signal of greening (fig. R4).

Figure. R4. Scatter plots between mean monthly $\frac{dLST_{bio}}{dLAI}$ and mean monthly $\frac{dLST_{bio}^{\alpha}}{dLAI}$ of from November to May

Considering albedo is not a climatic parameter, the bidirectional effect will not exist in the vegetation-albedo pathway. Meanwhile, the potential warming sensitivity induced by increased shortwave absorption is greater than the actual observed positive temperature sensitivity. These results suggest vegetation greening could significantly increase the shortwave radiation absorption and result in the warming effect of vegetation greening in boreal regions. In other words, the positive LST sensitivity in cold regions is not likely the result of two-way effects, or to say the least, the reversed temperature-vegetation effect does not play the dominant role.

In addition, we confirm the potential mechanism “increase in T_{air} can lead to a decrease in LAI in hotter regions” does not affect our result. We investigated the seasonal pattern of LST sensitivity in tropical regions (20°N to 20°S). If the mentioned mechanism is dominant, we expect a stronger negative sensitivity value in hotter months, as the higher temperature could threaten vegetation growth. As shown in Fig. R5, none of the above pattern is found in either the northern hemisphere tropics or southern hemisphere tropics. Thus, we argue that our sensitivities should not be affected by the bidirectional effects in hot regions.

Figure. R5. Seasonal pattern of LST sensitivity in tropical regions (20°S ~ 20°N)

Reference:

Li, Y., Zeng, Z., Huang, L., Lian, X., Piao, S., 2018. Comment on “Satellites reveal contrasting responses of regional climate to the widespread greening of Earth.” *Science* (80-.). 360, 1–3. <https://doi.org/10.1126/science.aap7950>

(2) The content of L98-118 is also similar to ref. 10. For example, Figure 1c with Table 1 of ref. 10.

Response: Thank you for pointing this out. The main reason that our result is similar to ref. 10 is that we use the same broad vegetation type classification scheme. The same scheme was used because the starting point of our research is to reconcile the disagreement between results from observation-based evidence and model-based simulation.

As expected, we find a comparable global mean sensitivity value ($-0.46 \text{ K}\cdot\text{m}^2\cdot\text{m}^{-2}$ vs $-0.36 \text{ K}\cdot\text{m}^2\cdot\text{m}^{-2}$). However, our results show much higher spatial standard variability (spatial standard deviation: $1.68 \text{ K}\cdot\text{m}^2\cdot\text{m}^{-2}$ vs $0.22 \text{ K}\cdot\text{m}^2\cdot\text{m}^{-2}$), and the mean sensitivity of the four vegetation types showed great differences. For instance, we find slight warming of forest greening globally, but ref. 10 indicates a cooling effect ($0.12 \text{ K}\cdot\text{m}^2\cdot\text{m}^{-2}$ vs $-0.23 \text{ K}\cdot\text{m}^2\cdot\text{m}^{-2}$); the sensitivities of the grasslands and croplands are 2 to 3 times higher than the results of ref. 10. Meanwhile, ref. 10 shows a cooling effect of greening in regions $>50^\circ\text{S/N}$, but our result shows a warming effect in high latitudes. Overall, our sensitivity results are more globally variable, while the LSM results (ref. 10) suggest a uniform global cooling effect with similar intensity.

To highlight the difference with the previous study, Fig.1c will be no longer the major result but moved to the supplementary (Supplementary Fig. 2), and we reduced the description (two paragraphs to one paragraph) of the variability of annual sensitivity with climate conditions and vegetation types in this section as follows (**Lines 71–102**):

Fig. 1 shows the annual biophysical sensitivity of LST to LAI ($\frac{dLST_{bio}}{dLAI}$) over the study period (2001–2018), which represents the potential annual mean temperature response to one LAI unit increase. Consistent with previous reports of the climate mitigation effect of earth greening (Forzieri et al., 2017; Zeng et al., 2018), we find that approximately 71% of the vegetated area shows negative sensitivity, and the global mean value is $-0.46 \pm 1.68 \text{ K}\cdot\text{m}^2\cdot\text{m}^{-2}$ (mean \pm spatial standard deviation) (Fig. 1a). Vegetation greening in dry and warm regions significantly cool the land surface, and this cooling diminishes and reverses to the warming effect with gradually decreased temperature or increased precipitation (Fig. 1b). Similar to the previous study of the biophysical impact of forestation on LST (Lawrence et al., 2022), this climatic variation could be further translated into the latitudinal dependence of a warming effect in northern high latitudes and cooling effects in other latitudes, with the transitional latitude near 50°N (Fig. 1c). However, the difference is found near the equator, where greening only induces a weak cooling effect, but previous studies suggest the strongest cooling of forestation here (Prevedello et al., 2019; Windisch et al., 2021). This difference is mainly due to the inherent difference between the abrupt change from openland to forest and the vegetation persistent greening. Apart from the impact of background climate, we further investigate the $\frac{dLST_{bio}}{dLAI}$ by aggregating all 12 IGBP vegetation types into four broad types (Supplementary Table 1), including forest, other wooded vegetation (OWV), grassland, and cropland (Supplementary Fig. 1). We find strong negative sensitivity in grasslands ($-0.94 \text{ K}\cdot\text{m}^2\cdot\text{m}^{-2}$) and croplands ($-0.83 \text{ K}\cdot\text{m}^2\cdot\text{m}^{-2}$), then followed by OWV ($-0.13 \text{ K}\cdot\text{m}^2\cdot\text{m}^{-2}$), and finally small positive sensitivity of forests ($0.16 \text{ K}\cdot\text{m}^2\cdot\text{m}^{-2}$) (Fig. 1c and Supplementary Fig. 2). Overall, comparing with the results from the controlled experiments of land surface model, the sensitivities estimated from our observational-based method show similar global magnitude, but with much larger spatial variability (Chen et al., 2020).

Figure 1. Mean annual biophysical sensitivity of LST to LAI ($\frac{dLST_{bio}}{dLAI}$) over the study period (2001–2018). a Spatial map of $\frac{dLST_{bio}}{dLAI}$. **b** Variation in sensitivity means across the climatic bins, in which $\frac{dLST_{bio}}{dLAI}$ is binned as a function of annual precipitation (P , x-axis) and air temperature (T_a , y-axis) from ERA land datasets. **c** Zonal means of $\frac{dLST_{bio}}{dLAI}$ across different broad vegetation types. The shaded area indicates the latitudinal standard deviation.

Reference:

Chen, C., Li, D., Li, Y., Piao, S., Wang, X., Huang, M., Gentine, P., Nemani, R.R., Myneni, R.B., Li1*, D., Li, Y., Piao, S., Wang, X., Huang, M., Gentine, P., Nemani, R.R., Myneni, R.B., 2020. Biophysical impacts of Earth greening largely controlled by aerodynamic resistance. *Sci. Adv.* 6, 1–10. <https://doi.org/10.1126/sciadv.abb1981>

Lawrence, D., Coe, M., Walker, W., Verchot, L., Vandecar, K., 2022. The Unseen Effects of Deforestation: Biophysical Effects on Climate. *Front. For. Glob. Chang.* 5, 1–13. <https://doi.org/10.3389/ffgc.2022.756115>

Piao, S., Wang, X., Park, T., Chen, C., Lian, X., He, Y., Bjerke, J.W., Chen, A., Ciais, P., Nemani, R.R., Myneni, R.B., 2020. Characteristics, drivers and feedbacks of global greening. *Nat. Rev. Earth Environ.*

Prevedello, J.A., Winck, G.R., Weber, M.M., Nichols, E., Sinervo, B., 2019. Impacts of forestation and deforestation on local temperature across the globe. *PLoS One* 14, 1–18. <https://doi.org/10.1371/journal.pone.0213368>

Windisch, M.G., Davin, E.L., Seneviratne, S.I., 2021. Prioritizing forestation based on biogeochemical and local biogeophysical impacts. *Nat. Clim. Chang.* 11, 867–871. <https://doi.org/10.1038/s41558-021-01161-z>

(3) In addition, talking about these sensitivities requires a stable background climate and vegetation growth (or very small changes). If there are significant changes in the background climate and vegetation growth, these sensitivities should also change. Therefore, the analysis in Section 3.3 is not convincing. A simple example is given to illustrate my concern.

Let $y = x^4 + z$, and assume z is independent of x .

Then, $dy/dx = 3x^3$;

When $x = 1$, $dy/dx = 3$;

When $x = 2$, $dy/dx = 24$;

The sensitivity of LST to LAI is similar to dy/dx , which changes if x is changing. You can only assume the sensitivity is a constant if x does not change much. Here y is analog to LST, and x is analog to LAI (x could also be the background climate).

Response: Thank you for your constructive comments. In the original manuscript, we simply assumed the mean value of multi-year sensitivities could be used to calculate the final LST signal. We fully agree with you that the sensitivity could change significantly due to the fluctuations of background climate and LAI. In the revised manuscript, we no longer use the multi-year mean sensitivity but the original monthly sensitivity to calculate the temperature effect. The specific modifications to the method are as follows (**Lines 657–674**):

The LST trend induced by greening (δLST_{bio}) over the study period (2001–2018) is the result of both biophysical sensitivity and LAI variation. Since the interannual variation of vegetation and climatic conditions can significantly affect the biophysical sensitivity, we used the original monthly sensitivity rather than the climatological value to estimate the temperature effect of greening. Specifically, all the sensitivity maps are firstly aggregated into 0.5° resolution to eliminate the impact of land cover or land management changes on LAI or sensitivity at the fine scale. Here, only grids with more than 50% valid values are retained. The Harmonic Analysis of Time Series (HANTS) was conducted to fill the invalid values of the sensitivity series for each 0.5° grid (Jiang et al., 2017). Then, the greening-induced LST variation (ΔLST_{bio}) can be calculated based on year-to-year LAI variation (ΔLAI) and the LST sensitivity ($\frac{dLST_{bio}}{dLAI}$) at monthly scale:

$$\Delta LST_{bio}(y, m) = \Delta LST_{bio}(y - 1, m) + \frac{\frac{dLST_{bio}}{dLAI}(y, m) + \frac{dLST_{bio}}{dLAI}(y - 1, m)}{2} \times \Delta LAI(y, m) \quad (12)$$

$$\Delta LAI(y, m) = LAI(y, m) - LAI(y - 1, m) \quad (13)$$

where y and m denote the year and month. Monthly δLST_{bio} could be computed by the linear trend of 17 years ΔLST_{bio} , and the annual or seasonal δLST_{bio} are similarly regressed from 17 years annual or seasonal ΔLST_{bio} (temporally aggregated from the monthly ΔLST_{bio}). The uncertainty of δLST_{bio} could be further estimated using the 95% confidence interval of the regression slope of year-to-year greening-induced temperature variation.

We find differences in the intensity of cooling globally and regionally after considering the interannual variation of LST sensitivity. For instance, δLST_{bio} decreased from 0.018 K/decade to 0.013 K/decade. Please refer to section “Climate benefit of greening over the last 20 years” for the changes in numerical results.

Meanwhile, we also made additional clarifications in section “Generation of climatological sensitivities” that the multi-year mean sensitivity is only used for analyzing the spatial and seasonal pattern (**Line 651–653**):

Notably, these climatology sensitivities are calculated for the spatial and seasonal patterns and the decomposition analysis, as they only represent the temperature response to greening under the multi-year average background climate.

Comments on the methodology:

There may be several problems with the methods section.

(1) First, I doubt that the statistical approach is able to decompose the multiple treatments involved in the surface energy balance, as I elaborated above.

Response: Thank you for raising this concern. Please refer to the explanation of the same background climate in our response to your major comments (2). Specifically, we assume all the pixels within the moving window share the same climate baseline. After the filter out the impact of vegetation type and topography, the differences in surface albedo and turbulence fluxes are driven by the LAI variation. Therefore, we argue that our spatial regression approach can extract the effect of LAI on the energy budget terms.

(2) I did not see a detailed description of the linear regression model.

Response: The equation of Theil-Sen's slope (Sen, 1968) was added in the revised manuscript (**Line 549**):

$$\text{slope} = \text{median} \left(\frac{y_i - y_j}{x_i - x_j} \right) \quad (1)$$

Reference:

Sen, P.K., 1968. Estimates of the Regression Coefficient Based on Kendall's Tau. *J. Am. Stat. Assoc.* 63, 1379–1389. <https://doi.org/10.1080/01621459.1968.10480934>

(3) L55: How exactly does the spatially moving window strategy work? It's too vague, more description is needed.

Response: Thank you for pointing this out. As formatting of Nature Communications requires a brief summary of our work in the instruction section, we cannot give more details about the method here. Therefore, we briefly state that our method is improved by the “space-for-time” approach here to acquire the LST sensitivity (**Lines 55–59**):

Due to the complicated bidirectional effect between vegetation growth and temperature variation, a spatial moving window strategy inspired by the ‘space-for-time approach is performed to exclude the impact of long-term climate signals on vegetation growth and acquire the LST sensitivity to LAI (Lee et al., 2011; Li et al., 2015).

In addition, we have rewritten the description of the spatial moving windows strategy in method section and it now reads (**Lines 536–556**):

This spatially moving window strategy is used to produce monthly $\frac{dLST_{bio}}{dLAI}$ over the study period. The specific way of the strategy works is as follows. For a given target pixel, all the potential samples for comparison are from the spatial nearby pixels within the moving window, which is set to 50×50 km (9×9 pixels at the equator) according to the previous studies (Ge et al., 2019; Zhao and Jackson, 2014). We further set two screening criteria for all candidates to exclude the influence of land cover and elevation difference: (1) the selected pixel should have the same main land cover type as the target pixel, and the coverage percentage difference should be less than 10% according to MODIS landcover data; (2) the elevation difference between the selected and target pixels should be less than 100 m. Then, we can obtain the biophysical sensitivity for the target pixel through the regression of LAI and LST differences between all selected comparison pixels and the target pixel. Here, the nonparametric Theil-Sen's slope is used to solve the potential skewed distribution problem of the samples (Sen, 1968):

$$\text{slope} = \text{median} \left(\frac{y_i - y_j}{x_i - x_j} \right) \quad (1)$$

Where x and y indicate the LAI and LST differences; i and j are the geolocations of samples within the moving window. Theil-Sen slope estimator adopts the median value of a range of possible slopes and is thus insensitive to the statistical outliers of the samples. In addition, we only calculate the sensitivity if there are at least four valid samples, and with a minimum LAI difference larger than $0.1 \text{ m}^2 \text{ m}^{-2}$ to further ensure the robustness of our result. Here, a positive sensitivity value means that vegetation greening has a warming effect on the local climate and vice versa.

Reference:

Ge, J., Guo, W., Pitman, A.J., de Kauwe, M.G., Chen, X., Fu, C., 2019. The nonradiative effect dominates local surface temperature change caused by afforestation in China. *J. Clim.* 32, 4445–4471. <https://doi.org/10.1175/JCLI-D-18-0772.1>

Lee, X., Goulden, M.L., Hollinger, D.Y., Barr, A., Black, T.A., Bohrer, G., Bracho, R., Drake, B., Goldstein, A., Gu, L., Katul, G., Kolb, T., Law, B.E., Margolis, H., Meyers, T., Monson, R., Munger, W., Oren, R., Paw U, K.T., Richardson, A.D., Schmid, H.P., Staebler, R., Wofsy, S., Zhao, L., 2011. Observed increase in local cooling effect of deforestation at higher latitudes. *Nature* 479, 384–387. <https://doi.org/10.1038/nature10588>

Li, Y., Zhao, M., Motesharrei, S., Mu, Q., Kalnay, E., Li, S., 2015. Local cooling and warming effects of forests based on satellite observations. *Nat. Commun.* 6, 1–10. <https://doi.org/10.1038/ncomms7603>

Sen, P.K., 1968. Estimates of the Regression Coefficient Based on Kendall's Tau. *J. Am. Stat. Assoc.* 63, 1379–1389. <https://doi.org/10.1080/01621459.1968.10480934>

Zhao, K., Jackson, R.B., 2014. Biophysical forcings of land-use changes from potential forestry activities in North America. *Ecol. Monogr.* 84, 329–353. <https://doi.org/10.1890/12-1705.1>

(4) I can't understand equations 2 and 3. They are not the standard equations for LE and H. How do you ensure that the turbulent flux changes in your analysis are caused by LAI changes and not by the energy closure preprocessing of these two equations?

Response: Thank you for raising these concerns. Equations 2 and 3 are the expressions of Bowen ratio correction method to distribute the energy closure residual. We perform this procedure because the decomposition model requires closure of surface energy balance. Here, we have rewritten the description of energy balance residual correction (**Line 572–585**):

To analyze the contribution of different biophysical feedbacks to the final LST sensitivity, we perform a decomposing procedure based on the energy balance equation Juang et al., 2007; Luysaert et al., 2014):

$$SW \downarrow (1 - \alpha) + \varepsilon LW \downarrow - \varepsilon \sigma (LST)^4 = H + LE + G \quad (2)$$

where $SW \downarrow$ and $LW \downarrow$ are shortwave and longwave downward radiation, α indicates albedo; σ denotes the Stephan-Boltzmann constant ($5.67 \times 10^{-8} \text{ W m}^{-2} \text{ K}^{-4}$); ε indicates surface emissivity, which is estimated from the empirical relationship with albedo for vegetated surfaces ($\varepsilon = 0.99 - 0.16\alpha$) (Juang et al., 2007); H , LE and G are respectively sensible heat, latent heat and ground heat flux. Notably, energy balance is required for the decomposition model (Liao et al., 2018), but the energy balance terms used in this study are from different datasets and thus are not closed. To account for this, we first perform the energy residual distribution using the Bowen ratio method, which assumes the ratio of H to LE is invariant (Jung et al., 2010).

$$LE_{correct} = (R_n - G) \times \frac{LE}{LE+H} \quad (3)$$

$$H_{correct} = (R_n - G) \times \frac{H}{LE+H} \quad (4)$$

$$R_n = SW \downarrow (1 - \alpha) + \varepsilon LW \downarrow - \varepsilon \sigma (LST)^4 \quad (5)$$

Moreover, we confirm that the energy closure process does not affect our conclusions. As shown in Fig. R6, the summed sensitivities from original turbulent fluxes data without energy closure shows larger absolute value and a similar latitudinal pattern compared with directly obtained LST sensitivity (dashed black line vs solid black line). They both demonstrate the larger negative sensitivity in mid-latitudes, and conversion of the relative dominance of radiative and non-radiative effects near 50°N, which corroborates the warming effect of boreal greening (left panel of Fig. R6). Second, our conclusions of the driving factors from sensitivities with or without correction are the same. Specifically, snow cover is found to drive the intensity of positive sensitivity in cold regions, as snow can intensify the radiative feedback rather than the non-radiative process (upper panel of Fig. R7); LAI affects non-radiative process through an exponential relationship and then drives the negative sensitivity in snow-free regions (lower panel of Fig. R7). In general, our closure treatment of the turbulent fluxes makes the sensitivity results from multi-source datasets

comparable in terms of the absolute value, but not produce misleading conclusions.

Figure. R6. Same with Fig. 3 in the main text, but the sensitivities of latent and sensible heat are not corrected by the energy closure.

Figure. R7. Same with Fig. 4 in the main text, but the sensitivities of latent and sensible heat are not corrected by the energy closure.

Notably, seasonal patterns of sensitivities from fluxes data with and without closure treatment in mid-latitudes are found different (right panel of Fig. R6 vs Fig. 3). The results from turbulent fluxes after the closure treatment are (Fig. 3) closer to the results of previous studies, which documents stronger turbulent flux sensitivity to LAI during the growing season (Forzieri et al., 2020). This suggests that our closure treatment can produce more reasonable results, and the reason behind might be the large uncertainty in the original turbulent flux products.

Reference:

Forzieri, G., Miralles, D.G., Ciais, P., Alkama, R., Ryu, Y., Duveiller, G., Zhang, K., Robertson, E., Kautz, M., Martens, B., Jiang, C., Arneth,

A., Georgievski, G., Li, W., Ceccherini, G., Anthoni, P., Lawrence, P., Wiltshire, A., Pongratz, J., Piao, S., Sitch, S., Goll, D.S., Arora, V.K., Lienert, S., Lombardozi, D., Kato, E., Nabel, J.E.M.S., Tian, H., Friedlingstein, P., Cescatti, A., 2020. Increased control of vegetation on global terrestrial energy fluxes. *Nat. Clim. Chang.* 10, 356–362.

Juang, J.Y., Katul, G., Siqueira, M., Stoy, P., Novick, K., 2007. Separating the effects of albedo from eco-physiological changes on surface temperature along a successional chronosequence in the southeastern United States. *Geophys. Res. Lett.* 34, 1–5. <https://doi.org/10.1029/2007GL031296>

Jung, M., Reichstein, M., Ciais, P., Seneviratne, S.I., Sheffield, J., Goulden, M.L., Bonan, G., Cescatti, A., Chen, J., De Jeu, R., Dolman, A.J., Eugster, W., Gerten, D., Gianelle, D., Gobron, N., Heinke, J., Kimball, J., Law, B.E., Montagnani, L., Mu, Q., Mueller, B., Oleson, K., Papale, D., Richardson, A.D., Rouspard, O., Running, S., Tomelleri, E., Viovy, N., Weber, U., Williams, C., Wood, E., Zaehle, S., Zhang, K., 2010. Recent decline in the global land evapotranspiration trend due to limited moisture supply. *Nature* 467, 951–954. <https://doi.org/10.1038/nature09396>

Liao, W., Rigden, A.J., Li, D., 2018. Attribution of Local Temperature Response to Deforestation. *J. Geophys. Res. Biogeosciences* 123, 1572–1587. <https://doi.org/10.1029/2018JG004401>

Luyssaert, S., Jammot, M., Stoy, P.C., Estel, S., Pongratz, J., Ceschia, E., Churkina, G., Don, A., Erb, K., Ferlicoq, M., Gielen, B., Grünwald, T., Houghton, R.A., Klumpp, K., Knohl, A., Kolb, T., Kuemmerle, T., Laurila, T., Lohila, A., Loustau, D., McGrath, M.J., Meyfroidt, P., Moors, E.J., Naudts, K., Novick, K., Otto, J., Pilegaard, K., Pio, C.A., Rambal, S., Reibmann, C., Ryder, J., Suyker, A.E., Varlagin, A., Wattenbach, M., Dolman, A.J., 2014. Land management and land-cover change have impacts of similar magnitude on surface temperature. *Nat. Clim. Chang.* 4, 389–393. <https://doi.org/10.1038/nclimate2196>

(5) Equations 6-10, why are total derivatives with respect to LAI? For example, there may be a variety of factors that affect changes in LE and H, such as wind speed, potential air temperature, changes in incident shortwave radiation, precipitation, soil moisture, VPD, etc. Assuming that LAI is the only driver of these surface changes is clearly invalid.

Response: Thank you for raising these important points. Indeed, turbulent fluxes are also affected by thermal and hydrological conditions of the lower atmosphere and soil. We argue these changes do not affect our results significantly due to the similar background climate assumption, as we mentioned in the response to your major comment (1). To further support our hypothesis, we further calculate the partial derivatives of LE to LAI ($\frac{\partial LE_{bio}}{\partial LAI}$), with other climatic variables (shortwave radiation, precipitation, surface air temperature, and wind speed from ERA Land datasets) included in the regression input. Similarly, the partial derivatives are obtained by the moving window strategies with the same model parameter (window size: 50×50 km; elevation difference < 100 m; main landcover fraction difference < 10%), and the multiple linear regression method is used to calculate the sensitivity. The comparison of climatological sensitivity of January and July is shown in Fig. R8, which shows good consistency between $\frac{\partial LE_{bio}}{\partial LAI}$ and $\frac{dLE_{bio}}{dLAI}$. This result indicates that the impact of meteorological factors on the energy budget terms is weakened due to the spatial moving window strategy.

Figure. R8. Latitudinal patterns comparison and density scatter plots between latent heat sensitivity from single LAI Theil-Sen's slope and multiple regression slope.

(6) L589-590: Why is it not induced by large-scale atmospheric circulation and biochemical processes? I cannot understand.

Response: Thank you for your attention to this issue. The main reason is the spatial regression method can only extract local effects. We have made an explanation in **Line 628–633**:

Theoretically, greening could also affect non-local climate, through horizontal heat or vapor transfer, and could also induce global cooling by absorbing the atmospheric CO₂. However, the radiation sensitivities from our method do not contain the contributions from these two processes, mainly because the spatial regression cannot extract the tele-connected or global uniform signal. In this study, the radiation sensitivity indicates the vertical impact of greening on climate. Specifically, greening directly alters energy absorption as well as water and heat exchange efficiency between the surface and atmosphere, subsequently affecting near-surface temperature, relative humidity, atmospheric transmittance and cloud cover locally (Duveiller et al., 2021), which could further alter surface downward shortwave and longwave radiation.

Reference:

Duveiller, G., Filipponi, F., Ceglar, A., Bojanowski, J., Alkama, R., Cescatti, A., 2021. Revealing the widespread potential of forests to increase low level cloud cover. *Nat. Commun.* 12, 1–15. <https://doi.org/10.1038/s41467-021-24551-5>

(7) Investigating seasonal energy balance should not ignore surface heat fluxes. They may have a significant contribution, especially in colder regions.

Response: Thank you for your valuable suggestions. We ignore the contribution of surface heat flux (G) partially because there are no proper G products to assess its contribution. Moreover, a few studies showed that vegetation changes have little impact on G on seasonal or interannual timescales (Juang et al., 2007; Lee et al., 2011; Lian et al., 2022).

Here, we performed the analysis of G using monthly simulations of NOAH land surface model in GLDAS

(Global Land Data Assimilation System) dataset, which has 0.25° spatial resolution. The same moving window strategy is used here to get the sensitivity of G to LAI, but the size is set to 3×3 pixels (about 70×70 km near the equator). As shown in Fig. R9a, we confirm the small contribution of G to the final LST sensitivity annually ($\frac{dLST_{bio}^G}{dLAI}$, $0.017 \text{ K m}^2 \text{ m}^{-2}$). The stronger impact of G is found in spring and autumn in mid-latitudes, but these temperature effects show different signs and offset each other. Similarly, $\frac{dLST_{bio}^G}{dLAI}$ for winter and summer also show a symmetrical pattern, leading to the final minor annual signal. Comparing with the final LST sensitivity or the sensitivity of albedo and turbulent fluxes, we confirm the small contribution of G at both annual and seasonal scales (Fig. R9b). Hence, we argue the effect of G is negligible in our attribution analysis.

Fig. R9 Latitudinal patterns of seasonal and annual $\frac{dLST_{bio}^G}{dLAI}$.

Reference:

- Juang, J.Y., Katul, G., Siqueira, M., Stoy, P., Novick, K., 2007. Separating the effects of albedo from eco-physiological changes on surface temperature along a successional chronosequence in the southeastern United States. *Geophys. Res. Lett.* 34, 1–5. <https://doi.org/10.1029/2007GL031296>
- Lee, X., Goulden, M.L., Hollinger, D.Y., Barr, A., Black, T.A., Bohrer, G., Bracho, R., Drake, B., Goldstein, A., Gu, L., Katul, G., Kolb, T., Law, B.E., Margolis, H., Meyers, T., Monson, R., Munger, W., Oren, R., Paw U, K.T., Richardson, A.D., Schmid, H.P., Staebler, R., Wofsy, S., Zhao, L., 2011. Observed increase in local cooling effect of deforestation at higher latitudes. *Nature* 479, 384–387. <https://doi.org/10.1038/nature10588>
- Lian, X., Jeong, S., Park, C., Xu, H., Li, L.Z.X., Wang, T., Gentine, P., Peñuelas, J., Piao, S., 2022. Biophysical impacts of northern vegetation changes on seasonal warming patterns. *Nat. Commun.* 13, 3925. <https://doi.org/10.1038/s41467-022-31671-z>

Comments specific to the line.

(1) L155: Which two processes?

Response: Thank you very much for pointing this out. We have revised the sentence and it now reads (**Line 171–174**):

Symmetrical latitudinal patterns are found between the radiative warming and non-radiative cooling, which suggests that their intensity may be controlled by the same factors.

(2) L170: Why are all changes in snow attributed to changes in LAI?

Response: Since the spatial nearby pixels share a similar background climate, we assume the snowfall and temperature of the comparison samples are the same. Hence, vegetation greening can affect the proportion

of snow exposed to direct sunlight, then affect the shortwave energy absorption. According to the Beer-Lambert Law, the energy that can be reflected by snow (I_{snow}) is related to LAI:

$$I_{snow} = I_0 \times \exp(-kLAI)$$

where I_0 is the downward solar radiation above the canopy and k is the extinction coefficient related to the distribution of leaf orientation within the canopy. The higher LAI value means the lower percentage of snow exposed to direct solar radiation, thus the lower shortwave radiation reflection from the hybrid pixel.

Reviewer #3 (Remarks to the Author):

Review of “Biophysical impacts of Earth greening can mitigate up to 50% of the land surface temperature warming”

This study applies an observationally based framework to quantify the temperature effects, globally and regionally, of the observed global greening trend over the past 20 years driven by biophysical mechanisms. Furthermore, the sensitivity of the temperature response is decomposed into contributions from radiative, non-radiative, and indirect climate feedbacks to delineate the underlying driving factors, as well as differences between vegetation types. The study generally confirms the main features of biophysical effects of vegetation changes found in previous work but adds value by using a purely observation-based approach and applying it to a less well studied case of greening rather than large-scale land use conversions. Overall, the paper is well structured with good figures. I find the study interesting, providing details on biome and process level that gives it the potential to be of importance for understanding the responses in further modeling-based work. Below are a few comments and questions for the authors' consideration before I can recommend publication, in particular related to treatment of uncertainties in the numbers and how the results are formulated in the context of mitigation.

Response: We appreciate your positive comments. We have revised our manuscript carefully and hope that the revised paper could be more suitable for publication. Specifically, the uncertainty estimation was added for the evaluated temperature effect of greening, and the discussions of our results were revised in the context of mitigation based on your comments. The point-to-point responses are as follows.

General comments:

(1) There is no uncertainty/error estimate in any of the numbers provided, whereas Fig. 1 show substantial standard errors, in some cases even in terms of sign sensitivity with vegetation type. This should preferably be quantified and at the very least better discussed, including in the abstract.

Response: Thank you very much for your valuable comments and constructive suggestions. In fact, the larger standard deviation presented in the sensitivity is the result of the spatial variability of sensitivity. Meanwhile, we fully agree that uncertainty estimation is essential for the temperature trends induced by greening (δLST_{bio}). In the new version, we have changed the method for calculating δLST_{bio} based on the major comments (3) of reviewer #2. The description of uncertainty calculation based on the new method is as follows (**Lines 673–674**):

The uncertainty of δLST_{bio} could be further estimated using the 95% confidence interval of the regression slope of year-to-year greening-induced temperature variation.

In addition, the uncertainty of mitigation ratios is also quantified as follows (**Line 693–696**):

Then, the mitigation percentage can be represented by the ratio of greening induced temperature trend to the warming rate without the mitigation of greening $\left(-\frac{\delta LST_{bio}}{\delta T_{obs}-\delta LST_{bio}} \times 100\%\right)$. Furthermore, the uncertainty of this ratio is calculated based on the error propagation of δT_{obs} and δLST_{bio} .

All the numbers of δLST_{bio} in abstract and main text are updated with uncertainty estimations. Here, we show an example of the number modification (**Line 260–263**):

Correspondingly, the global mean greening-induced temperature trend (δLST_{bio}) is -0.013 ± 0.009 K/decade (Fig. 5a), which offset 4.6 ± 3.2 % of surface warming trend of 0.289 K/decade within the vegetated

area.

(2) It is not immediately clear to me if the percentage numbers given are the fraction of the total temperature impact due to greening only that is “mitigated” or the fraction of the total observed warming (i.e. total human-induced). Should be clearly written in the text throughout.

Response: The fraction number is closer to your latter definition. We rewrite the method for calculating this ratio to avoid ambiguity (**Line 690–696**):

Here, δT_{obs} represent the temperature response to all forcings, which is dominated by the increasing CO_2 concentration induced warming, and we use $\delta T_{obs} - \delta LST_{bio}$ to represent the temperature trend without biophysical feedback of greening. Then, the mitigation percentage can be represented by the ratio of greening induced temperature trend to the warming rate without the mitigation of greening $\left(-\frac{\delta LST_{bio}}{\delta T_{obs} - \delta LST_{bio}} \times 100\%\right)$. Meanwhile, the uncertainty of this ratio is further calculated based on the error propagation of δT_{obs} and δLST_{bio} .

Here is an example, if the greening could induce a cooling of -0.2 K/decade and the observed warming is 0.2 K/decade, we can say that greening mitigate $-\frac{-0.2 \text{ K/decade}}{(0.2 - (-0.2)) \text{ K/decade}} \times 100\% = 50\%$ of the warming.

(3) The title gives impression of a very large mitigation potential of the greening. Upon seeing the findings, I feel it should be explicit about this being local/regional effects (and depending on the answer to the comment above, maybe further revised). Later parts of the paper states that the global magnitude of biophysically based mitigation is limited.

Response: Thank you for pointing this out. Indeed, the large mitigation potential of greening is only found regionally. We have changed the title into:

Biophysical Impacts of Earth Greening Can Mitigate Substantial Regional Land Surface Temperature Warming

(4) The results are generally discussed in terms of mitigation and the authors for instance say that the cooling effect derived over India and China can mitigate up to 50% of global warming. But how confident are the authors in the potential for extrapolating these results based on historically observed changes? I.e. does dynamic vegetation changes or saturation effects come into play? Strictly, one would only be able to state that these changes have offset some of the total warming, which maybe should be reflected in the wording.

Response: Thank you for your attention to this issue. In this paper, the temperature effect induced vegetation change is solved by our spatial regression method. Unfortunately, this effect is difficult to be extrapolated directly from historically observed changes, since the actual observed changes of the climate contains complex signal from natural and anthropogenic forcings.

Our assessment represents only the climate effects of vegetation greening over 2001 to 2018. Since the vegetation greening trend is only confirmed after the 1980s, the dynamic vegetation changes may have insignificant impact on the temperature in terms of long-term historical climate change. Meanwhile, our results show that the temperature trend induced by greening is influenced by saturation effects, especially in forests with high LAI values.

In addition, we have revised the verb “mitigate” before the ratio with “offset” as you suggested.

Nevertheless, we fully understand your concerns about the powerful regional mitigation effects of greening. Here, we should emphasize that the results are based on the land surface temperature (LST), which is different from the air temperature used in the field of climate change. In fact, there could be discrepancies in temperature sensitivity if different temperature measurements are used. A discussion of the potential overestimation of the mitigation effect due to the temperature measurements can be found in result section **(Line 432–442)**:

However, when the acquisition of sensitivity is not limited to our method, previous studies have also shown the different responses of air temperature and surface temperature to vegetation changes (Novick and Katul, 2020; Winckler et al., 2019). Specifically, based on the ESM simulations of different scenarios, the local 2 m air temperature sensitivity is about 35% to 65% of the local surface temperature sensitivity. If we take this difference in temperature measurements into account, the cooling effect of greening will be also correspondingly halved. However, the climate mitigation ratio could reach 24.5% and 10.5% in India and China (assuming the air temperature sensitivity is half the surface temperature sensitivity), which still shows the strong climate mitigation effects of anthropogenic greening. Thus, we argue the role of vegetation greening cannot be ignored in the assessment of future climate, especially for the hot spots of greening induced directly by human activities.

Reference:

Novick, K.A., Katul, G.G., 2020. The Duality of Reforestation Impacts on Surface and Air Temperature. *J. Geophys. Res. Biogeosciences* 125, 1–15. <https://doi.org/10.1029/2019JG005543>

Winckler, J., Reick, C.H., Luysaert, S., Cescatti, A., Stoy, P.C., Lejeune, Q., Raddatz, T., Chlond, A., Heidkamp, M., Pongratz, J., 2019. Different response of surface temperature and air temperature to deforestation in climate models. *Earth Syst. Dyn.* 10, 473–484. <https://doi.org/10.5194/esd-10-473-2019>

(5) Related: from previous studies, it is my impression that the observed global greening is primarily driven by fertilization effects due to increased CO₂, which makes it more a consequence of our emissions. How actively can we continue to enhance the greening without exacerbating the CO₂ or other pollution? Should perhaps a mention of the driving factors of greening be included? Some discussion of irrigation and forestation is there, but these are the types of conversions that the authors explicitly say they do not address here...

Response: Thank you for your valuable suggestions. Earth greening since the 1980s is mainly due to the increasing CO₂, climate change, and direct human land management. In the recent 20 years, the greening trend is found stronger in China and India, where human activity could be the major driving factor of greening. This direct human land management and land use (such as agricultural intensification and ecological projects) can be the solution for enhancing the greening without exacerbating the CO₂ or other pollution. In the revised manuscript, the driving factors of greening including increasing CO₂ and human activities are discussed as follows **(Lines 375–389)**:

The increasing CO₂ concentration is the major driving force of both global warming and earth greening in recent decades. However, compared with the global warming rate, we confirm the very limited effect of biophysical-based climate mitigation from earth greening. Meanwhile, our results also provide evidence of the significant surface cooling in regions with extensive greening trends. Specifically, we find that greening-related cooling can offset about 20% and 40% of the warming trend in China and India (Fig. 6), respectively. These results highlight the role of the biophysical impact of greening in future adaptation strategies against

ongoing warming. A previous study proved that China and India are the leaders in global greening in the 21st century, which is achieved by human land-use management, such as the afforestation project in China (Ge et al., 2019) and the increased harvested area by fertilization and irrigation in India (Chen et al., 2019). This suggests the large potential of human land use and land management strategies and ecological projects to mitigate climate pressure, not only through carbon uptake from the atmosphere but also the biophysical processes.

About the type conversion, we admit it is incorrect to separate the contribution of LULCC to vegetation greening. In the revised manuscript, those regions with landcover changes are also included in the analysis through a spatial aggregation strategy (please refer to the response to comment (5) of reviewer #1).

Reference:

Chen, C., Park, T., Wang, X., Piao, S., Xu, B., Chaturvedi, R.K., Fuchs, R., Brovkin, V., Ciais, P., Fensholt, R., Tømmervik, H., Bala, G., Zhu, Z., Nemani, R.R., Myneni, R.B., 2019. China and India lead in greening of the world through land-use management. *Nat. Sustain.* 2, 122–129. <https://doi.org/10.1038/s41893-019-0220-7>

Ge, J., Guo, W., Pitman, A.J., de Kauwe, M.G., Chen, X., Fu, C., 2019. The nonradiative effect dominates local surface temperature change caused by afforestation in China. *J. Clim.* 32, 4445–4471. <https://doi.org/10.1175/JCLI-D-18-0772.1>

(6) Is LAI a sufficient variable? How about SAI – would that factor into the, in particular, albedo changes? Moreover, there are a number of satellite-derived products out there, with large discrepancies suggested by some. A word on the GLASS LAI compared to other products would be useful.

Response: Thank you for your constructive comments. We choose LAI here, but not other spectral vegetation indexes (such as NDVI and EVI) here, because LAI is a vegetation structural variable with explicit physical meaning. More importantly, LAI is an input driving factor of land surface models, which means our results could be comparable with LSM.

We perform additional sensitivity tests using different LAI products, please refer to the response to comment (2) of reviewer #1.

Specific:

(1) Line 36: LUCC – should it rather be the more commonly used LULCC?

Response: We have revised this abbreviation as suggested (**Line 39 and 524**).

(2) Line 64: this could be a good place to clarify what temperature signals are compared, i.e. is this the total observed temperature change relative to some baseline climatology or the total temperature signal due to greening alone?

Response: Thank you for your suggestion. We have revised the sentence and it now reads (**Line 66–68**):

This estimated signal is subsequently compared with the observed historical temperature variation to evaluate the potential climate benefits of greening at global and regional scales.

(3) Line 87: seems like a significant difference - can the authors offer some suggestions for why?

Response: Thank you for your comment. We have explained the reason for this difference in **Line 92–93**:

This difference is mainly due to the inherent difference between the abrupt change from openland to forest and the vegetation persistent greening.

More specifically, the temperature effect of afforestation can be recognized as the integral of the sensitivity ($\int_a^b \frac{dLST_{bio}}{dLAI} dLAI$). Here, a and b indicate the LAI value before and after the forestation. However, the sensitivity indicates only derivative of LST over LAI values, that is, the temperature response to a slight disturbance to LAI value.

(4) Line 243: Could be useful with a figure that show the regional ratio of biophysically-driven to total (as in the current Fig. S9) temperature

Response: Thanks for your suggestion. We present the ratio in the way you propose (Fig. 6a) and the discussion is as follows (Line 309–317):

To investigate the regional climate benefit of greening, we calculate the mitigation ratio of greening using observed surface air temperature trend (δT_{obs}) and the estimated greening-induced temperature trend (δLST_{bio}) at pixel and national scale (see method). The higher value of this ratio indicates larger warming trend the biophysical feedback of greening can offset. As shown in Fig. 6a, regions where greening can significantly mitigate climate change are overlapped with significant greening areas, (e.g., China, India, Europe, southern Brazil, and the central United States). These regions have also been confirmed by the previous study to dominate the global greening signal after the 21st century. (Chen et al., 2019).

Figure 6. Potential mitigation effect of biophysical impact of earth greening. **a** Spatial map of mitigation ratio. Areas with statistically significant LAI trends are masked by black dots (Mann-Kendall test, $P < 0.05$) Only pixels that are significant at 95% confidence interval are shown. **b** Comparison of observed LST trend and greening-induced LST in 10 countries with a sizeable vegetated area. Error bars indicate the uncertainty of greening induced temperature trend. The percentage under each blue bar indicates the climate mitigation ratio (mean \pm uncertainty). Here, European Union (EU) is included in the analysis. Other abbreviations: US, United States; DRC, Democratic Republic of the Congo.

(5) Line 245: does this also affect the total temperature impact of the greening and hence the ratio/fraction of warming that is offset?

Response: Indeed, the cooling effect of those areas with significant greening does affect the total temperature impact and the ratio. Here, we made more discussion to show the strong contribution of such regions (**Line 263–267**):

When considering those pixels with statistically significant LAI trends (Mann-Kendall test, $P < 0.05$), the cooling trend can increase to -0.029 ± 0.008 K/decade. Greening in such regions could offset about $9.2 \pm 3.7\%$ of the corresponding warming trend, and contribute about 68% of the global cooling signal.

(6) Throughout: some typos that need fixing...

Response: We apologize for the mistakes in the manuscript. We have checked the paper carefully and corrected typos and grammar errors.

REVIEWERS' COMMENTS

Reviewer #1 (Remarks to the Author):

The authors have addressed my concerns in the revised version and I therefore recommend the publication of this work.

Reviewer #2 (Remarks to the Author):

I thank the authors for writing a perfect rebuttal, and I enjoyed reading it. I realized that the authors are very thoughtful about this study. The current version has been greatly improved and is in a good shape. Excellent Job! I recommend publishing. No need for me to review it again.

I only have some minor comments:

Perhaps this paper is helpful. One seasonal pattern analysis based on model simulations was published in Nature Communications several months ago. Please see <https://www.nature.com/articles/s41467-022-31671-z>.

In rebuttal #4, how robust is it to assume that the ratio of H to LE is invariant? Vegetation changes will lead to changes in surface biophysical factors (e.g., roughness, stomatal conductance), and once these surface biophysical factors change, the Bowen ratio should change. LST in many areas, especially during the growing season, should be governed by changes caused by these surface biophysical factors, through turbulence fluxes (more important than albedo). It is helpful to the readers if the authors could add some discussion on this hypothesis.

Reviewer #3 (Remarks to the Author):

The authors have addressed at least my concerns and comments in a satisfactory manner. Hence, pending that the other reviewers don't find new issues introduced by the revisions, I would support publication of the manuscript.

I have only one additional small comment: in the new methodology description on line 583-585 in the track changes version (Eq. 3-5), it's not entirely clear to me what is meant by the subscript "correct" and how these terms enter into later equations. (Also, the sentence before say "we first perform (...) " which seems to apply that there should be some sort of "we then" - either I'm missing something or there is something missing...)

Response to Reviewers' Comments

We greatly appreciate the positive comments from the reviewers. Below are the point-by-point responses to the comments, along with the revision of the manuscript (typed in italics) and the location of the revision (typed in bold). Hope the revision will make it more acceptable for publication.

Reviewer #1 (Remarks to the Author):

The authors have addressed my concerns in the revised version and I therefore recommend the publication of this work.

Response: We appreciate all the constructive comments by reviewer #1, which helped a lot for the improvement of the manuscript.

Reviewer #2 (Remarks to the Author):

I thank the authors for writing a perfect rebuttal, and I enjoyed reading it. I realized that the authors are very thoughtful about this study. The current version has been greatly improved and is in a good shape. Excellent Job! I recommend publishing. No need for me to review it again.

Response: Thank you for the valuable comments for improving the quality of the manuscript.

I only have some minor comments:

Perhaps this paper is helpful. One seasonal pattern analysis based on model simulations was published in Nature Communications several months ago. Please see <https://www.nature.com/articles/s41467-022-31671-z>.

Response: Thank you for pointing this. We have cited the Lian et al. (2022) recommended by the reviewer in the revised manuscript (**Line 667-668**):

Lian, X. et al. Biophysical impacts of northern vegetation changes on seasonal warming patterns. Nat. Commun. 13, 3925 (2022).

In rebuttal #4, how robust is it to assume that the ratio of H to LE is invariant? Vegetation changes will lead to changes in surface biophysical factors (e.g., roughness, stomatal conductance), and once these surface biophysical factors change, the Bowen ratio should change. LST in many areas, especially during the growing season, should be governed by changes caused by these surface biophysical factors, through turbulence fluxes (more important than albedo). It is helpful to the readers if the authors could add some discussion on this hypothesis.

Response: Thank for your comment. Indeed, vegetation changes affect LST mainly through modifying the Bowen ratio. However, the residual distribution process based on the invariant Bowen ratio assumption is only used to force the energy closure of multi-source energy balance terms, which does not mean that the Bowen ratio is constant when the LAI value has changed. We have added the explanation to this issue in **Line 473**:

Notably, this invariant assumption does not mean the Bowen ratio is constant when the vegetation has changed.

Reviewer #3 (Remarks to the Author):

The authors have addressed at least my concerns and comments in a satisfactory manner. Hence, pending that the other reviewers don't find new issues introduced by the revisions, I would support

publication of the manuscript.

Response: Thank you for the constructive comments for in the last round review, which helped a lot for the improvement of the manuscript.

I have only one additional small comment: in the new methodology description on line 583-585 in the track changes version (Eq. 3-5), it's not entirely clear to me what is meant by the subscript "correct" and how these terms enter into later equations. (Also, the sentence before say "we first perform (...)" which seems to apply that there should be some sort of "we then" - either I'm missing something or there is something missing...)

Response: Thank you for pointing this out. The subscript "correct" was changed to "corrected" (**Line 470-471**):

$$LE_{corrected} = (R_n - G) \times \frac{LE}{LE+H} \quad (3)$$

$$H_{corrected} = (R_n - G) \times \frac{H}{LE+H} \quad (4)$$

Meanwhile, we used the word "then" instead of "first", as suggested by the reviewer (**Line 468**).

In addition, we added the sentence in **Line 474** to illustrate how the corrected turbulent fluxes data are used: *Subsequently, the corrected turbulent fluxes ($LE_{corrected}$ and $H_{corrected}$) data are then used for the further attribution analysis in this paper.*